# Coresets for Robust Training of Neural Networks against Noisy Labels

**Baharan Mirzasoleiman**
Department of Computer Science
University of California Los Angeles
Los Angeles, CA
baharan@cs.ucla.edu

**Kaidi Cao**
Department of Computer Science
Stanford University
Stanford, CA
kaidicao@cs.stanford.edu

**Jure Leskovec**
Department of Computer Science
Stanford University
Stanford, CA
jure@cs.stanford.edu

## Abstract

Modern neural networks have the capacity to overfit noisy labels frequently found in real-world datasets. Although great progress has been made, existing techniques are limited in providing theoretical guarantees for the performance of the neural networks trained with noisy labels. Here we propose a novel approach with strong theoretical guarantees for robust training of deep networks trained with noisy labels. The key idea behind our method is to select weighted subsets (coresets) of clean data points that provide an approximately low-rank Jacobian matrix. We then prove that gradient descent applied to the subsets do not overfit the noisy labels. Our extensive experiments corroborate our theory and demonstrate that deep networks trained on our subsets achieve a significantly superior performance compared to state-of-the art, e.g., 6% increase in accuracy on CIFAR-10 with 80% noisy labels, and 7% increase in accuracy on mini Webvision[1].

## 1 Introduction

The success of deep neural networks relies heavily on the quality of training data, and in particular accurate labels of the training examples. However, maintaining label quality becomes very expensive for large datasets, and hence mislabeled data points are ubiquitous in large real-world datasets [21]. As deep neural networks have the capacity to essentially memorize any (even random) labeling of the data [49], noisy labels have a drastic effect on the generalization performance of deep neural networks. Therefore, it becomes crucial to develop methods with strong theoretical guarantees for robust training of neural networks against noisy labels. Such guarantees become of the utmost importance in safety-critical systems, such as aircraft, autonomous cars, and medical devices.

There has been a great empirical progress in robust training of neural networks against noisy labels. Existing directions mainly focus on: estimating the noise transition matrix [13, 34], designing robust loss functions [12, 42, 44, 52], correcting noisy labels [26, 36, 41], using explicit regularization techniques [7, 50, 51], and selecting or reweighting training examples [9, 14, 17, 27, 37, 44]. In general, estimating the noise transition matrix is challenging, correcting noisy labels is vulnerable to overfitting, and designing robust loss functions or using explicit regularization cannot achieve

state-of-the-art performance [16, 23, 51]. Therefore, the most promising methods rely on selecting or reweighting training examples by knowledge distillation from auxiliary models [14, 17, 27], or exploiting an extra clean labelled dataset containing no noisy labels [17, 25, 37, 43, 50]. In practice, training reliable auxiliary models could be challenging, and relying on an extra dataset is restrictive as it requires the training and extra dataset to follow the same distribution. Nevertheless, the major limitation of the state-of-the-art methods is their inability to provide theoretical guarantees for the performance of neural networks trained with noisy labels.

There has been a few recent efforts to theoretically explain the effectiveness of regularization and early stopping in generalization of over-parameterized neural networks trained on noisy labels [16, 23]. Specifically, Hu et al. [16] proved that when width of the hidden layers is sufficiently large (polynomial in the size of the training data), gradient descent with regularization by distance to initialization corresponds to kernel ridge regression using the Neural Tangent Kernel (NTK). Kernel ridge regression performs comparably to early-stopped gradient descent [35, 45], and leads to a generalization guarantee in presence of noisy labels. In another work, Li et al. [23] proved that under a rich (clusterable) dataset model, a one-hidden layer neural network trained with gradient descent first fits the correct labels, and then starts to overfit the noisy labels. This is consistent with the previous empirical findings showing that deep networks tend to learn simple examples first, then gradually memorize harder instances [5]. In practice, however, regularization and early-stopping provide robustness only under relatively low levels of noise (up to 20% of noisy labels) [16, 23].

Here we develop a principled technique, CRUST, with strong theoretical guarantees for robust training of neural networks against noisy labels. The key idea of our method is to carefully select subsets of *clean* data points that allow the neural network to effectively learn from the training data, but prevent it to overfit noisy labels. To find such subsets, we rely on recent results that characterize the training dynamics based on properties of neural network Jacobian matrix containing all its first-order partial derivatives. In particular, (1) learning along prominent singular vectors of the Jacobian is fast and generalizes well, while learning along small singular vectors is slow and leads to overfitting; and (2) label noise falls on the space of small singular values and impedes generalization [32]. To effectively and robustly learn from the training data, CRUST efficiently finds subsets of clean and diverse data points for which the neural network has an approximately low-rank Jacobian matrix.

We show that the set of *medoids* of data points in the gradient space that minimizes the average gradient dissimilarity to all the other data points satisfies the above properties. To avoid overfitting noisy labels, CRUST iteratively extracts and trains on the set of updated medoids. We prove that for large enough coresets and a constant fraction of noisy labels, deep networks trained with gradient descent on the medoids found by CRUST do not overfit the noisy labels. We then explain how mixing up [51] the centers with a few other data points reduces the error of gradient descent updates on the coresets. Effectively, clean coresets found by CRUST improve the generalization performance by reducing the ratio of noisy labels and their alignment with the space of small singular values.

We conduct experiments on noisy versions of CIFAR-10 and CIFAR-100 [22] with noisy labels generated by random flipping the original ones, and the mini Webvision datasets [24] which is a benchmark consisting of images crawled from websites, containing real-world noisy labels. Empirical results demonstrate that the robustness of deep models trained by CRUST is superior to state-of-the-art baselines, e.g. 6% increase in accuracy on CIFAR-10 with 80% noisy labels, and 7% increase in accuracy on mini Webvision. We note that CRUST achieves state-of-the-art performance without the need for training any auxiliary model or utilizing an extra clean dataset.

## 2   Additional Related Work

In practice, deeper and wider neural networks generalize better [40, 48]. Theoretically, recent results proved that when the number of hidden nodes is polynomial in the size of the dataset, neural network parameters stay close to their initialization, where the training landscape is almost linear [1, 4, 8, 11], or convex and semi-smooth [2]. For such networks, (stochastic) gradient descent with random initialization can almost always drive the training loss to 0, and overfit any (random or noisy) labeling of the data. Importantly, these results utilize the property that the Jacobian of the neural network is well-conditioned at a random initialization if the dataset is sufficiently diverse.

More closely related to our work is the recent result of [32] which proved that along the directions associated with large singular values of a neural network Jacobian, learning is fast and generalizes well. In contrast, early stopping can help with generalization along directions associated with small

singular values. This is consistent with prior results proving the effectiveness of regularization and early stopping for providing robustness against noisy labels [16, 23]. These results, however, are restricted to unrealistically wide networks, and in practice are only effective under low levels of noise.

On the other hand, our method, CRUST, provides rigorous guarantees for robust training of *arbitrary* deep neural networks against noisy labels, by efficiently extracting subsets of clean data points that provide an approximately low-rank Jacobian matrix during the training. Effectively, the extracted subsets do not allow the network to overfit noise, and hence CRUST can quickly train a model that generalizes well. Unlike existing analytical results that are limited to a neighborhood around a random initialization, our method captures the change in Jacobian structure of deep networks for arbitrary parameter values during training. As a result, it achieves state-of-the-art performance both under mild as well as severe noise.

## 3 Problem Setting: Learning from Noisy Labeled Data

In this section we formally describe the problem of learning from datasets with noisy labels. Suppose we have a dataset $\mathcal{D} = \{(\boldsymbol{x}_i, y_i)\}_{i=1}^n \subset \mathbb{R}^d \times \mathbb{R}$, where $(\boldsymbol{x}_i, y_i)$ denotes the $i$-th sample with input $\boldsymbol{x}_i \in \mathbb{R}^d$ and its observed label $y_i \in \mathbb{R}$. We assume that the labels $\{y_i\}_{i=1}^n$ belong to one of $C$ classes. Specifically, $y_i \in \{\nu_1, \nu_2, \cdots, \nu_C\}$ with $\{\nu_j\}_{j=1}^C \in [-1, 1]$. We further assume that the labels are separated with margin $\delta \le |\nu_r - \nu_s|$ for all $r, s \in [C], r \ne s$. Suppose we only observe inputs and their noisy labels $\{y_i\}_{i=1}^n$, but do not observe true labels $\{\tilde{y}_i\}_{i=1}^n$. For each class $1 \le j \le C$, a fraction of the labels associated with that class are assigned to another label chosen from $\{\nu_j\}_{j=1}^C$.

Let $f(\boldsymbol{W}, \boldsymbol{x})$ be an $L$-layer fully connected neural network with scalar output, where $\boldsymbol{x} \in \mathbb{R}^d$ is the input and $\boldsymbol{W} = (\boldsymbol{W}^{(1)}, \cdots, \boldsymbol{W}^{(L)})$ is all the network parameters. Here, $\boldsymbol{W}^{(l)} \in \mathbb{R}^{d_l \times d_{l-1}}$ is the weight matrix in the $l$-th layer ($d_0 = d, d_L = 1$). For simplicity, we assume all the parameters are aggregated in a vector, i.e., $\boldsymbol{W} \in \mathbb{R}^m$, where $m = \sum_{l=2}^L d_l \times d_{l-1}$. Suppose that the network is trained by minimizing the squared loss over the noisy training dataset $\mathcal{D} = \{(\boldsymbol{x}_i, y_i)\}_{i=1}^n$:

$$\mathcal{L}(\boldsymbol{W}) = \frac{1}{2} \sum_{i \in V} (y_i - f(\boldsymbol{W}, \boldsymbol{x}_i))^2, \tag{1}$$

where $V = \{1, \cdots, n\}$ is the set of all training examples. We apply gradient descent with a constant learning rate $\eta$, starting from an initial point $\boldsymbol{W}^0$ to minimize $\mathcal{L}(\boldsymbol{W})$. The iterations take the form

$$\boldsymbol{W}^{\tau+1} = \boldsymbol{W}^\tau - \eta \nabla \mathcal{L}(\boldsymbol{W}^\tau, \boldsymbol{X}), \quad \nabla \mathcal{L}(\boldsymbol{W}, \boldsymbol{X}) = \mathcal{J}^T(\boldsymbol{W}, \boldsymbol{X})(f(\boldsymbol{W}, \boldsymbol{X}) - \boldsymbol{y}), \tag{2}$$

where $\mathcal{J}(\boldsymbol{W}, \boldsymbol{X}) \in \mathbb{R}^{n \times m}$ is the Jacobian matrix associated with the nonlinear mapping $f$ defined as

$$\mathcal{J}(\boldsymbol{W}, \boldsymbol{X}) = \Big[ \frac{\partial f(\boldsymbol{W}, \boldsymbol{x}_1)}{\partial \boldsymbol{W}} \ \cdots \ \frac{\partial f(\boldsymbol{W}, \boldsymbol{x}_n)}{\partial \boldsymbol{W}} \Big]^T. \tag{3}$$

The goal is to learn a function $f : \mathbb{R}^d \to \mathbb{R}$ (in the form of a neural network) that can predict the true labels $\{\tilde{y}_i\}_{i=1}^n$ on the dataset $\mathcal{D}$. In the rest of the paper, we use $\boldsymbol{X} = (\boldsymbol{x}_1, \cdots, \boldsymbol{x}_n), \boldsymbol{y} = (y_1, \cdots, y_n)^T, \tilde{\boldsymbol{y}} = (\tilde{y}_1, \cdots, \tilde{y}_n)^T$. Furthermore, we use $\boldsymbol{X}_S, \boldsymbol{y}_S$, and $\mathcal{J}(\boldsymbol{W}, X_S)$ to denote inputs, labels, and the Jacobian matrix associated with elements in a subset $S \subseteq V$ of data points, respectively.

## 4 Our Approach: CRUST

In this section we present our main results. We first introduce our method CRUST that selects subsets of *clean* data points that has an approximately low-rank Jacobian and do not allow gradient descent to overfit noisy labels. Then, we show how mixing up the subsets with a few other data points can further reduce the error of gradient descent updates, and improve the generalization performance.

### 4.1 Extracting Clean Subsets with Approximately Low-rank Jacobian

The key idea of our method is to carefully select subsets of data points that allow the neural network to effectively learn from the clean training data, but prevent it to overfit noisy labels. Recent result on optimization and generalization of neural networks show that the Jacobian of typical neural networks exhibits an approximately low-rank structure, i.e., a number of singular values are large and the

remaining majority of the spectrum consists of small singular values. Consequently, the Jacobian spectrum can be split into *information* space $\mathcal{I}$, and *nuisance* space $\mathcal{N}$, associated with the large and small singular values [32]. Formally, for complementary subspaces $\mathcal{S}_-, \mathcal{S}_+ \subset \mathbb{R}^n$, and for all unit norm vectors $\boldsymbol{v} \in \mathcal{S}_+$, $\boldsymbol{w} \in \mathcal{S}_-$, and scalars $0 \le \mu \ll \alpha \le \beta$, we have

$$\alpha \le \|\mathcal{J}^T(\boldsymbol{W}, \boldsymbol{X})\boldsymbol{v}\|_2 \le \beta, \quad \text{and} \quad \|\mathcal{J}^T(\boldsymbol{W}, \boldsymbol{X})\boldsymbol{w}\|_2 \le \mu. \tag{4}$$

There are two key observations [23, 32]: (1) While learning over the low-dimensional information space is fast and generalizes well, learning over the high-dimensional nuisance space is slow and leads to overfitting; and (2) The generalization capability and dynamics of training is dictated by how well the label, $\boldsymbol{y}$, and residual vector, $\boldsymbol{r} = f(\boldsymbol{W}, \boldsymbol{X}) - \boldsymbol{y}$, are aligned with the information space. If the residual vector is very well aligned with the singular vectors associated with the top singular values of $\mathcal{J}(\boldsymbol{W}, \boldsymbol{X})$, the gradient update, $\nabla \mathcal{L}(\boldsymbol{W}) = \mathcal{J}^T(\boldsymbol{W}, \boldsymbol{X})\boldsymbol{r}$, significantly reduces the misfit allowing substantial reduction in the training error. Importantly, the residual and label of most of the clean data points fall on the information space, while the residual and label of noisy data points fall on the nuisance space and impede training and generalization.

To avoid overfitting noisy labels, one can leverage the first observation above, and iteratively selects subsets $S$ of $k$ data points that provide the best rank-$k$ approximation to the Jacobian matrix. In doing so, gradient descent applied to the subsets cannot overfit the noisy labels. Formally:

$$S^*(\boldsymbol{W}) = \arg\min_{S \subseteq V} \|\mathcal{J}^T(\boldsymbol{W}, \boldsymbol{X}) - P_S \mathcal{J}^T(\boldsymbol{W}, \boldsymbol{X})\|_2 \quad \text{s.t.} \quad |S| \le k, \tag{5}$$

where $\mathcal{J}(\boldsymbol{W}, \boldsymbol{X}_S) \in \mathbb{R}^{k \times m}$ is the set of $k$ rows of the Jacobian matrix associated to $\boldsymbol{X}_S$, and $P_S = \mathcal{J}^T(\boldsymbol{W}, \boldsymbol{X}_S)\mathcal{J}(\boldsymbol{W}, \boldsymbol{X}_S)$ denotes the projection onto the $k$-dimensional space spanned by the rows of $\mathcal{J}(\boldsymbol{W}, \boldsymbol{X}_S)$. Existing techniques to find the best subset of $k$ rows or columns from an $n \times m$ matrix have a computational complexity of $poly(n, m, k)$ [3, 10, 20], where $n, m$ are the number of data points and parameters in the network. Note that the subset $S^*(\boldsymbol{W})$ depends on the parameter vector $\boldsymbol{W}$ and a new subset should be extracted after every parameter update. Furthermore, calculating the Jacobian matrix requires backpropagation on the entire dataset which could be very expensive for deep networks. Therefore, the computational complexity of the above methods becomes prohibitive for over-parameterized neural networks trained on large datasets. Most importantly, while this approach prohibits overfitting, it does not help identifying the clean data points.

To achieve a good generalization performance, our approach takes advantage of both the above mentioned observations. In particular, our goal is to find representative subsets of $k$ diverse data points with clean labels that span the information space $\mathcal{I}$, and provide an approximately low-rank Jacobian matrix. The important observation is that as nuisance space is very high dimensional, data points with noisy labels spread out in the *gradient* space. In contrast, information space is low-dimensional and data points with clean labels that have similar gradients cluster closely together. The set of most centrally located clean data points in the gradient space can be found by solving the following $k$-medoids problem:

$$S^*(\boldsymbol{W}) \in \arg\min_{S \subseteq V} \sum_{i \in V} \min_{j \in S} d_{ij}(\boldsymbol{W}) \quad \text{s.t.} \quad |S| \le k, \tag{6}$$

where $d_{ij}(\boldsymbol{W}) = \|\nabla \mathcal{L}(\boldsymbol{W}, \boldsymbol{x}_i) - \nabla \mathcal{L}(\boldsymbol{W}, \boldsymbol{x}_j)\|_2$ is the pairwise dissimilarity between gradients of data points $i$ and $j$. Note that the above formulation does not provide the best rank-$k$ approximation of the Jacobian matrix. However, as the $k$-medoids objective selects a diverse set of clean data points, the minimum singular value of the Jacobian of the selected subset projected over the subspace $\mathcal{S}_+$, i.e., $\sigma_{\min}(\mathcal{J}(\boldsymbol{W}, \boldsymbol{X}_S^*), \mathcal{S}_+)$, will be large. Next, we weight the derivative of every medoid $j \in S^*$ by the size of its corresponding cluster $r_j = \sum_{i \in V} \mathbb{1}[j = \arg\min_{s \in S^*} d_{is}]$ to create the weighted Jacobian matrix $\mathcal{J}_r(\boldsymbol{W}, \boldsymbol{X}_{S^*}) = \text{diag}([r_1, \cdots, r_k])\mathcal{J}(\boldsymbol{W}, \boldsymbol{X}_{S^*}) \in \mathbb{R}^{k \times m}$. We can establish the following upper and lower bounds on the singular values $\sigma_{i \in [k]}(\mathcal{J}_r(\boldsymbol{W}, \boldsymbol{X}_{S^*}), \mathcal{S}_+)$ of the weighted Jacobian over $\mathcal{S}_+$:

$$\sqrt{r_{\min}}\sigma_{\min}(\mathcal{J}(\boldsymbol{W}, \boldsymbol{X}_{S^*}), \mathcal{S}_+) \le \sigma_{i \in [k]}(\mathcal{J}_r(\boldsymbol{W}, \boldsymbol{X}_{S^*}), \mathcal{S}_+) \le \sqrt{r_{\max}}\|\mathcal{J}(\boldsymbol{W}, \boldsymbol{X}_{S^*})\|, \tag{7}$$

where $r_{\min} = \min_{j \in [k]} r_j$ and $r_{\max} = \max_{j \in [k]} r_j$, and we get an error of $\epsilon$ in approximating the largest singular value of the neural network Jacobian, $\epsilon \le |\sqrt{r_{\max}}\|\mathcal{J}(\boldsymbol{W}, \boldsymbol{X}_{S^*})\| - \|\mathcal{J}(\boldsymbol{W}, \boldsymbol{X})\||$. Now, we apply gradient descent updates in Eq. (2) to the weighted Jacobian $\mathcal{J}_r(\boldsymbol{W}, \boldsymbol{X}_{S^*})$ of the $k$ extracted medoids:

$$\boldsymbol{W}^{\tau+1} = \boldsymbol{W}^\tau - \eta \mathcal{J}_r^T(\boldsymbol{W}, \boldsymbol{X}_{S^*})(f(\boldsymbol{W}, \boldsymbol{X}_{S^*}) - \boldsymbol{y}_{S^*}), \tag{8}$$

Note that we still need backpropagation on the entire dataset to be able to compute pairwise dissimilarities $d_{ij}$. For neural networks, it is shown that the variation of the gradient norms is mostly captured

by the gradient of the loss w.r.t. the input to the last layer of the network [19]. This argument can be used to efficiently upper-bound the normed difference between pairwise gradient dissimilarities [30]:

$$d_{ij}(\boldsymbol{W}) = \|\nabla\mathcal{L}(\boldsymbol{W}, \boldsymbol{x}_i) - \nabla\mathcal{L}(\boldsymbol{W}, \boldsymbol{x}_j)\|_2 \le c_1 \|\Sigma'_L(\boldsymbol{z}_i^L)\nabla_i^L\mathcal{L} - \Sigma'_L(\boldsymbol{z}_j^L)\nabla_j^L\mathcal{L}\|_2 + c_2, \quad (9)$$

where $\Sigma'_L(\boldsymbol{z}_i^L)\nabla_i^L\mathcal{L}$ is gradient of the loss function $\mathcal{L}$ w.r.t. the input to the last layer $L$ for data point $i$, and $c_2, c_2$ are constants. The above upper-bound is marginally more expensive to calculate than the value of the loss since it can be computed in a closed form in terms of $z^L$. Hence, $d_{ij}^u = \|\Sigma'_L(\boldsymbol{z}_i^L)\nabla_i^L\mathcal{L} - \Sigma'_L(\boldsymbol{z}_j^L)\nabla_j^L\mathcal{L}\|_2$ can be efficiently calculated. We note that although the upper-bounds $d_{ij}^u$ have a lower dimensionality than $d_{ij}$, noisy data points still spread out in this lower-dimensional space, and hence are not selected as medoids. This is confirmed by out experiments (Fig. 1 (a)).

Having upper-bounds on the pairwise gradient dissimilarities, we can efficiently find a near-optimal solution for problem (6) by turning it into a *submodular maximization* problem. A set function $F : 2^V \to \mathbb{R}^+$ is submodular if $F(S \cup \{e\}) - F(S) \ge F(T \cup \{e\}) - F(T)$, for any $S \subseteq T \subseteq V$ and $e \in V \setminus T$. $F$ is *monotone* if $F(e|S) \ge 0$ for any $e \in V \setminus S$ and $S \subseteq V$. Minimizing the objective in Problem (6) is equivalent to maximizing the following submodular facility location function:

$$S^*(\boldsymbol{W}) \in \arg\max_{\substack{S \subseteq V, \\ |S| \le k}} F(S, \boldsymbol{W}), \qquad F(S, \boldsymbol{W}) = \sum_{i \in V} \max_{j \in S} d_0 - d'_{ij}(\boldsymbol{W}), \quad (10)$$

where $d_0$ is a constant satisfying $d_0 \ge d_{ij}^u(\boldsymbol{W})$, for all $i, j \in V$. For maximizing the above monotone submodular function, the classical greedy algorithm provides a constant $(1 - 1/e)$-approximation. The greedy algorithm starts with the empty set $S_0 = \varnothing$, and at each iteration $t$, it chooses an element $e \in V$ that maximizes the marginal utility $F(e|S_t) = F(S_t \cup \{e\}) - F(S_t)$. Formally, $S_t = S_{t-1} \cup \{\arg\max_{e \in V} F(e|S_{t-1})\}$. The computational complexity of the greedy algorithm is $\mathcal{O}(nk)$. However, its complexity can be reduced to $\mathcal{O}(|V|)$ using stochastic methods [29], and can be further improved using lazy evaluation [28] and distributed implementations [31]. Note that this complexity does not involve any backpropagation as we use the upper-bounds calculated in Eq. (9). Hence, the subsets can be found very efficiently, in parallel from all classes. Unlike majority of the existing techniques for robust training against noisy labels that has a large computational complexity, robust training with CRUST is even faster than training on the entire dataset. We also note that Problem (10) can be addressed in the streaming scenario for very large datasets [6].

Our experiments confirm that CRUST can successfully find almost all the clean data points (*c.f.* Fig. 1 (a)). The following theorem guarantees that for a small fraction $\rho$ of noisy labels in the selected subsets, deep networks trained with gradient descent do not overfit the noisy labels.

**Theorem 4.1** *Assume that we apply gradient descent on the least-squares loss in Eq. (2) to train a neural network on a dataset with noisy labels. Furthermore, suppose that the Jacobian mapping is $L$-smooth[2]. Assume that the dataset has a label margin of $\delta$, and coresets found by CRUST contain a fraction of $\rho < \delta/8$ noisy labels. If the coresets approximate the Jacobian matrix by an error of at most $\epsilon \le \mathcal{O}(\frac{\delta\alpha^2}{k\beta\log(\sqrt{k}/\rho)})$, where $\alpha = \sqrt{r_{\min}}\sigma_{\min}(\mathcal{J}(\boldsymbol{W}, \boldsymbol{X}_S)), \beta = \|\mathcal{J}(\boldsymbol{W}, \boldsymbol{X})\| + \epsilon$, then for $L \le \frac{\alpha\beta}{L\sqrt{2k}}$ and step size $\eta = \frac{1}{2\beta^2}$, after $\tau \ge \mathcal{O}(\frac{1}{\eta\alpha^2}\log(\frac{\sqrt{n}}{\rho}))$ iterations the network classifies all the selected elements correctly.*

The proof can be found in the Appendix. Note that the elements of the selected subsets are mostly clean, and hence the noise ratio $\rho$ is much smaller in the subsets compared to the entire dataset. Very recently, [32] showed that the classification error of neural networks trained on noisy datasets of size $n$ is controlled by the portion of the labels that fall over the nuisance space, i.e., $\|\Pi_{\mathcal{N}}(\boldsymbol{y})\|/\sqrt{n}$. Coresets of size $k$ selected by CRUST are mostly clean. For such subsets, the label vector is mostly aligned with the information space, and thus $\|\Pi_{\mathcal{N}}(\boldsymbol{y}_S)\|/\sqrt{k}$ is smaller. Our method improves the generalization performance by extracting subsets $S$ of size $k$ for which $\|\Pi_{\mathcal{N}}(\boldsymbol{y}_S)\|/\sqrt{k} \le \|\Pi_{\mathcal{N}}(\boldsymbol{y})\|/\sqrt{n}$. While defer the formal generalization proof to future work, our experiments show that even under severe noise (80% noisy labels), CRUST successfully finds the clean data points and achieves a superior generalization performance (*c.f.* Fig. 1).

Next, we discuss how to reduce the error of backpropagation on the weighted centers.

**Algorithm 1** CORESETS FOR ROBUST TRAINING AGAINST NOISY LABELS (CRUST)

**Input:** The noisy set $\mathcal{D} = \{(\boldsymbol{x}_i, y_i)\}_{i=1}^{n}$, number of iterations $T$.
**Output:** Output model parameters $\boldsymbol{W}^T$.

1: **for** $\tau = 1, \cdots, T$ **do**
2:     $S^\tau = \varnothing$.
3:     **for** $c \in \{1, \cdots, C\}$ **do**
4:         $U_c^\tau = \{(\boldsymbol{x}_i, y_i) \in \mathcal{D} | f(\boldsymbol{W}^\tau, \boldsymbol{x}_i) = \nu_c\}$, $n_c = |U_c^\tau|/n$.     ▷ Classify based on predictions.
5:         $d_{ij}^u$ =upper-bounded pairwise gradient dissimilarities for $i, j \in U_c^\tau$         ▷ Eq. 9.
6:         $S_c^\tau = \{k \cdot n_c\text{-medoids from } U_c^\tau \text{ using } d_{ij}^u\}$         ▷ The greedy algorithm.
7:         **for** $j \in S_c^\tau$ **do**
8:             $V_j^\tau = \{i \in U_c^\tau | j = \arg\min_{v \in S_c^\tau} d_{iv}^u\}$
9:             $R_j^\tau$ = small random sample from $V_j^\tau$.
10:            $\hat{D}_j^\tau$ =Mixup $(\boldsymbol{x}_j, y_j)$ with $\{(\boldsymbol{x}_i, y_i) \mid i \in R_j^\tau\}$         ▷ Eq. (11).
11:            $r_i = |V_j^\tau|/|R_j^\tau|, \ \forall i \in R_j^\tau$         ▷ Coreset weights in Eq. (8).
12:            $S^\tau = S^\tau \cup \hat{D}_j^\tau$
13:        **end for**
14:    **end for**
15:    Update the parameters $\boldsymbol{W}^\tau$ using weighted gradient descent on $S^\tau$.         ▷ Eq. (2).
16: **end for**

## 4.2 Further Reducing the Error of Coresets

There are two potential sources of error during weighted gradient descent updates in Eq. (8). First, we have an $\epsilon$ error in estimating the prominent singular value of the Jacobian matrix. And second, although the $k$-medoids formulation selects centers of clustered clean data points in the gradient space, there is still a small chance, in particular early in training process when the gradients are more uniformly distributed, that the coresets contain some noisy labels. Both errors can be alleviated if we slightly relax the constraint of training on *exact* feature vectors and their labels, and allow training on combinations of every center $j \in S$ with a few examples in its corresponding cluster $V_j$.

This is the idea behind mixup [51]. It extends the training distribution with convex combinations of pairs of examples and their labels. For every cluster $V_j$, we select a few data points $R_j \subset V_j \setminus \{j\}, |R_j| \ll |V_j|$ uniformly at random, and for every data point $(\boldsymbol{x}_i, y_i), i \in R_j$ we mix it up with the corresponding center $(\boldsymbol{x}_j, y_j), j \in S^*$, to get the set $\hat{D}_j$ of mixed up points:

$$\hat{D}_j = \{(\hat{\boldsymbol{x}}, \hat{y}) \mid \hat{x} = \lambda \boldsymbol{x}_i + (1 - \lambda)\boldsymbol{x}_j, \ \hat{y} = \lambda y_i + (1 - \lambda)y_j \quad \forall i \in R_j\}, \tag{11}$$

where $\lambda \sim \text{Beta}(\alpha, \alpha) \in [0, 1]$ and $\alpha \in \mathbb{R}^+$. The mixup hyper-parameter $\alpha$ controls the strength of interpolation between feature-label pairs. Our experiments show that the subsets chosen by CRUST contain mostly clean data points, but may contain some noisy labels early during the training (Fig. 1 (a)). Mixing up the centers with as few as one example from the corresponding cluster, i.e. $|R_j| = 1$, can reduce the effect of potential noisy labels in the selected subsets. Therefore, mixup can further help improving the generalization performance, as is confirmed by our experiments (Table 2).

## 4.3 Iteratively Reducing Noise

The subsets found in Problem (10) depend on the parameter vector $\boldsymbol{W}$ and need to be updated during the training. Let $\boldsymbol{W}^\tau$ be the parameter vector at iteration $\tau$. At update time $\tau \in [T]$, we first classify data points based on the updated predictions $\boldsymbol{y}^\tau = f(\boldsymbol{W}^\tau, \boldsymbol{X})$. We denote by $U_c^\tau = \{(\boldsymbol{x}_i, y_i) \in \mathcal{D} | f(\boldsymbol{W}^\tau, \boldsymbol{x}_i) = \nu_c\}$ the set of data points labeled as $\nu_c$ in iteration $\tau$. Then, we find $S(\boldsymbol{W}^\tau)$ by greedily extracting $k \cdot n_c$ medoids from each class, where $n_c = |U_c|/n$ is the fraction of data points in class $c \in [C]$. Finding separate coresets from each class can further help to not cluster together noisy data points spread out in the nuisance space, and improves the accuracy of the extracted coresets. Next, we partition the data points in every class to by assigning every data point to its closest medoid. Formally, for partition $V_j^\tau$ we have $V_j^\tau = \{i \in U_c^\tau | j = \arg\min_{j \in S^\tau} d_{ij}^u\}$. Finally, we take a small random sample $R_j^\tau$ from every partition $V_j^\tau$, and for every data point $i \in R_j^\tau$ we mix it up with the corresponding medoid $j \in S(\boldsymbol{W}^\tau)$ according to Eq. (11), and add the generated set $\hat{D}_j^\tau$ of mixed up data points to the training set. In our experiments we use $|R_j^\tau| = 1$. At update time $\tau$,

Table 1: Average test accuracy (5 runs) on CIFAR-10 and CIFAR-100. The best test accuracy is marked in bold. CRUST achieves up to 6% improvement (3.15% in average) over the strongest baseline INCV. We note the superior performance of CRUST under 80% label noise.

| Dataset | CIFAR-10 | | | | CIFAR-100 | | |
|---|---|---|---|---|---|---|---|
| Noise Type | Sym | | | Asym | Sym | | Asym |
| Noise Ratio | 20 | 50 | 80 | 40 | 20 | 50 | 40 |
| F-correction | $85.1 \pm 0.4$ | $76.0 \pm 0.2$ | $34.8 \pm 4.5$ | $83.6 \pm 2.2$ | $55.8 \pm 0.5$ | $43.3 \pm 0.7$ | $42.3 \pm 0.7$ |
| Decoupling | $86.7 \pm 0.3$ | $79.3 \pm 0.6$ | $36.9 \pm 4.6$ | $75.3 \pm 0.8$ | $57.6 \pm 0.5$ | $45.7 \pm 0.4$ | $43.1 \pm 0.4$ |
| Co-teaching | $89.1 \pm 0.3$ | $82.1 \pm 0.6$ | $16.2 \pm 3.2$ | $84.6 \pm 2.8$ | $64.0 \pm 0.3$ | $52.3 \pm 0.4$ | $47.7 \pm 1.2$ |
| MentorNet | $88.4 \pm 0.5$ | $77.1 \pm 0.4$ | $28.9 \pm 2.3$ | $77.3 \pm 0.8$ | $63.0 \pm 0.4$ | $46.4 \pm 0.4$ | $42.4 \pm 0.5$ |
| D2L | $86.1 \pm 0.4$ | $67.4 \pm 3.6$ | $10.0 \pm 0.1$ | $85.6 \pm 1.2$ | $12.5 \pm 4.2$ | $5.6 \pm 5.4$ | $14.1 \pm 5.8$ |
| INCV | $89.7 \pm 0.2$ | $84.8 \pm 0.3$ | $52.3 \pm 3.5$ | $86.0 \pm 0.5$ | $60.2 \pm 0.2$ | $53.1 \pm 0.4$ | $50.7 \pm 0.2$ |
| T-Revision | $79.3 \pm 0.5$ | $78.5 \pm 0.6$ | $36.2 \pm 1.6$ | $76.3 \pm 0.8$ | $52.4 \pm 0.3$ | $37.6 \pm 0.3$ | $32.3 \pm 0.4$ |
| L_DMI | $84.3 \pm 0.4$ | $78.8 \pm 0.5$ | $20.9 \pm 2.2$ | $84.8 \pm 0.7$ | $56.8 \pm 0.4$ | $42.2 \pm 0.5$ | $39.5 \pm 0.4$ |
| CRUST | $\mathbf{91.1 \pm 0.2}$ | $\mathbf{86.3 \pm 0.3}$ | $\mathbf{58.3 \pm 1.8}$ | $\mathbf{88.8 \pm 0.4}$ | $\mathbf{65.2 \pm 0.2}$ | $\mathbf{56.4 \pm 0.4}$ | $\mathbf{53.0 \pm 0.2}$ |

we train on the union of sets generated by mixup, i.e. $D^\tau = \{\hat{D}_1^\tau \cup \cdots \cup \hat{D}_k^\tau\}$, where every data point $i \in \hat{D}_j^\tau$ is weighted by $r_i = |V_j^\tau|/|R_j^\tau|$. The pseudo code of CRUST is given in Algorithm 1.

Note that while CRUST updates the coreset during the training, as almost all the clean data points are contained in the coresets in every iteration, gradient descent can successfully contract their residuals in Theorem 4.1 and fit the correct labels. Since CRUST finds a new subset at every iteration $\tau$, we need to use $\alpha = \min_\tau \sqrt{r_{\min}^\tau} \sigma_{\min}(\mathcal{J}(\boldsymbol{W}^\tau, \boldsymbol{X}_{S^\tau}))$, and $\beta = \max_\tau \sqrt{r_{\max}^\tau} \|\mathcal{J}(\boldsymbol{W}^\tau, \boldsymbol{X}_{S^\tau})\|$ in Theorem 4.1, where $\alpha$ and $\beta$ are the minimum and maximum singular values of the Jacobian of the subsets weighted by $\boldsymbol{r}^\tau$ found by CRUST, during $T$ steps of gradient descent updates.

# 5 Experiments

We evaluate our method on artificially corrupted versions of CIFAR-10 and CIFAR-100 [22] with controllable degrees of label noise, as well as a real-world large-scale dataset mini WebVision [24], which contains real noisy labels. Our algorithm is developed with PyTorch [33]. We use 1 Nvidia GTX 1080 Ti for all CIFAR experiments and 4 for training on the mini WebVision dataset.

**Baselines.** We compare our approach with multiple state-of-the-art methods for robust training against label corruption. (1) F-correction [34] first naively trains a neural network using ERM, then estimates the noise transition matrix $T$. $T$ is then used to construct a corrected loss function with which the model will be retrained. (2) MentorNet [17] first pretrains a teacher network to mimic a curriculum. The student network is then trained with the sample reweighting scheme provided by the teacher network. (4) D2L [26] reduces the effect of noisy labels on learning the true data distribution after learning rate annealing using corrected labels. (3) Decoupling [27] trains two networks simultaneously, and the two networks only train on a subset of samples that do not have the same prediction in every mini batch. (5) Co-teaching [14] also maintains two networks in the training time. Each network selects clean data (samples with small loss) and guide the other network to train on its selected clean subset. (6) INCV [9] first estimates the noise transition matrix $T$ through cross validation, then applies iterative Co-teaching by including samples with small losses. (7) T-Revision [46] designs a deep-learning-based risk-consistent estimator to tune the transition matrix accurately. (8) L_DMI [47] proposes information theoretic noise-robust loss function based on generalized mutual information.

## 5.1 Empirical results on artificially corrupted CIFAR

We first evaluate our method on CIFAR-10 and CIFAR-100, which contain 50,000 training images and 10,000 test images of size $32 \times 32$ with 10 and 100 classes, respectively. We follow testing protocol adopted in [9, 14], by considering both symmetric and asymmetric label noise. Specifically, we test noise ratio of 0.2, 0.5, 0.8 for symmetric noise, and 0.4 for asymmetric noise.

In our experiments, we train ResNet-32 [15] for 120 epochs with a minibatch of 128. We use SGD with an initial learning rate of 0.1 and decays at epoch 80, 100 by a factor of 10 to optimize

Table 2: Ablation study on CIFAR-10 with 20% and 50% symmetric noise. ✓ indicates the corresponding component. coreset w/label and coreset w/label correspond to finding coresets separately from every class based on their observed noisy labels, or labels predicted by the model being trained.

| Component | | | | Noise Ratio | |
|---|---|---|---|---|---|
| coreset w/ label | coreset w/ pred. | w/o mixup | w/ mixup | 20 | 50 |
| ✓ | | ✓ | | 90.21 | 84.92 |
| ✓ | | | ✓ | 90.48 | 85.23 |
| | ✓ | ✓ | | 90.71 | 85.57 |
| | ✓ | | ✓ | 91.12 | 86.27 |

the objective, with a momentum of 0.9 and weight decay of $5 \times 10^{-4}$. We only use simple data augmentation following [15]: we first pad 4 pixels on every side of the image, and then randomly crop a $32 \times 32$ image from the padded image. We flip the image horizontally with a probability of 0.5. For CRUST, we select coresets of size 50% of the size of the dataset unless otherwise stated.

We report top-1 test accuracy of various methods in Table 1. Our proposed method CRUST outperforms all the baselines in terms of average test accuracy. While INCV attempts to find a subset of the training set with heuristics, our theoretically-principled method can successfully distinguish data points with correct labels from those with noisy labels, which results in a clear improvement across all different settings. It can be seen that CRUST achieves a consistent improvement by an average of 3.15%, under various symmetric and asymmetric noisy scenarios, compared to the strongest baseline INCV. Interestingly, CRUST achieves the largest improvement of 6% over INCV, under sever 80% label noise. This shows the effectiveness of our method in extracting data points with clean labels from a large number of noisy data points, compared to other baselines.

## 5.2 Ablation study on each component

Here we investigate the effect of each component of CRUST and its importance, for robust training on CIFAR-10 with 20% and 50% symmetric noise. Table 2 summarizes the results.

**Effect of the coreset.** Based on the empirical results, the coreset plays an important role in improving the generalization of the trained networks. It is worthwhile noticing that by greedily finding the coreset based on the gradients, we can outperform INCV already. This clearly corroborate our theory and shows the effectiveness of CRUST in filtering the noise and extracting data points with correct labels, compared to other heuristics. It also confirms our argument that although upper-bounded gradient dissimilarities in Eq. (9) has a much lower dimensionality compared to the exact gradients, noisy data points still spread out in the gradient space. Therefore, CRUST can successfully identify central data points with clean labels in the gradient space.

**Effect of mixup and model update.** As discussed in Sec. 4.2, mixup can reduce the bias of estimating the full gradient with the coreset. Moreover, finding separate coresets from each class can further help filtering noisy labels, and improves the accuracy of the extracted coresets. At the beginning of every epoch, CRUST updates the predictions based on the current model parameters and extract corsets from every class separately. We observed that both components, namely mixup and extracting coresets separately from each class based on the predictions of the model being trained, further improve the generalization and hence the final accuracy.

**Size of the coresets.** Fig. 1 demonstrates training curve for CIFAR-10 with 50% symmetric noise. Fig. 1(a) shows the accuracy of coresets of size 30%, 50%, and 70% selected by CRUST. We observe that for various sizes of coresets, the number of noisy centers decreases over time. Furthermore, the fraction of correct labels in the coresets (label accuracy) decreases when the size of the selected centers increases from 30% to 70%. This demonstrates that CRUST identifies clean data points first. Fig. 1 (b), (c) show the train and test accuracy, when training with CRUST on coresets of various sizes. We can see that coresets of size 30% achieve a lower accuracy as they are too small to accurately estimate the spectrum of the information space and achieve a good generalization performance. Coresets of size 70% achieve a lower training and test accuracy compared to coresets of size 50%. As 50% of the labels are noisy, subsets of size 70% contain at least 20% noisy labels and the model eventually overfits the noise.

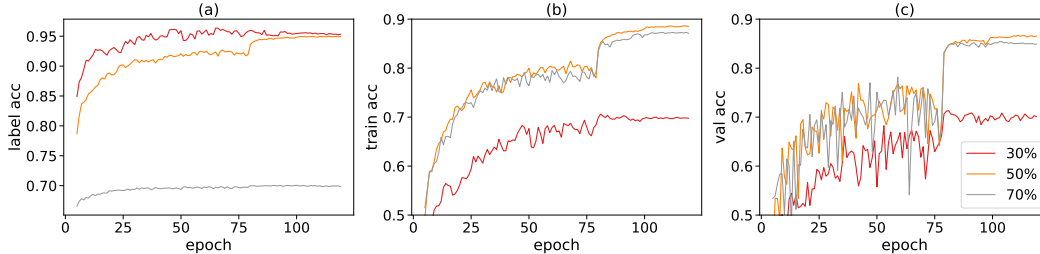

Figure 1: Training curve for CIFAR-10 with 50% symmetric noise. (a) The fraction of correct labels in the coreset (label accuracy) with respect to epochs. (b) The accuracy on the whole training set with respect to epochs. (c) The accuracy on test set with respect to epochs. Here we consider coresets of size 30%, 50%, and 70% of the CIFAR-10 training dataset.

Table 3: Test accuracy on mini WebVision. The best test accuracy is marked in bold. CRUST achieves up to 7.16% improvement (6.46% in average) in top-1 accuracy over the strongest baseline INCV.

| Method | WebVision | | ImageNet | |
|---|---|---|---|---|
| | Top-1 | Top-5 | Top-1 | Top-5 |
| F-correction | 61.12 | 82.68 | 57.36 | 82.36 |
| Decoupling | 62.54 | 84.74 | 58.26 | 82.26 |
| Co-teaching | 63.58 | 85.20 | 61.48 | 84.70 |
| MentorNet | 63.00 | 81.40 | 57.80 | 79.92 |
| D2L | 62.68 | 84.00 | 57.80 | 81.36 |
| INCV | 65.24 | 85.34 | 61.60 | 84.98 |
| **CRUST** | **72.40** | **89.56** | **67.36** | **87.84** |

## 5.3 Empirical results on mini WebVision

WebVision is real-world dataset with inherent noisy labels [24]. It contains 2.4 million images crawled from Google and Flickr that share the same 1000 classes from the ImageNet dataset. The noise ratio in classes varies from 0.5% to 88% (Fig. 4 in [24] shows the noise distribution). We follow the setting in [17] and create a mini WebVision that consists of the top 50 classes in the Google subset with 66,000 images. We use both WebVision and ImageNet test sets for testing the performance of the model trained on coresets of size 50% of the data found by CRUST. We train InceptionResNet-v2 [39] for 90 epochs with a starting learning rate of 0.1. We anneal the learning rate at epoch 30 and 60, respectively. The results are shown in Table 3. It can be seen that our method consistently outperforms other baselines, and achieves an average of 5% improvement in the test accuracy, compared to INCV.

## 6 Conclusion

We proposed a novel approach with strong theoretical guarantees for robust training of neural networks against noisy labels. Our method, CRUST, relies on the following key observations: (1) Learning along prominent singular vectors of the Jacobian is fast and generalizes well, while learning along small singular vectors is slow and leads to overfitting; and (2) The generalization capability and dynamics of training is dictated by how well the label and residual vector are aligned with the information space. To achieve a good generalization performance and avoid overfitting, CRUST iteratively selects subsets of clean data points that provide an approximately low-rank Jacobian matrix. We proved that for a constant fraction of noisy labels in the subsets, neural networks trained with gradient descent applied to the subsets found by CRUST correctly classify all its data points. At the same time, our method improves the generalization performance of the deep network by decreasing the portion of noisy labels that fall over the nuisance space of the network Jacobian. Our extensive experiments demonstrated the effectiveness of our method in providing robustness against noisy labels. In particular, we showed that deep networks trained on the our subsets achieve a significantly superior performance, e.g., 6% increase in accuracy on CIFAR-10 with 80% noisy labels, and 7% increase in accuracy on mini Webvision, compared to state-of-the-art baselines.

## Broader Impact

Deep neural networks achieve impressive results in a wide variety of domains, including vision and speech recognition. The quality of the trained deep models on such datasets increases logarithmically with the size of the data [38]. This improvement, however, is contingent on the availability of reliable and accurate labels. In practice, collecting large high quality datasets is often very expensive and time-consuming. For example, labeling of medical images depends on domain experts and hence is very resource-intensive. In some applications, it necessitate obtaining consensus labels or labels from multiple experts and methods for aggregating those annotations to get the ground truth labels [18]. In some domains, crowd-sourcing methods are used to obtain labels from non-experts. An alternative solution is automated mining of data, e.g., from the Internet by using different image-level tags that can be regarded as labels. These solutions are cheaper and more time-efficient than human annotations, but label noise in such datasets is expected to be higher than in expert-labeled datasets. Noisy labels have a drastic effect on the generalization performance of deep neural networks. This prevents deep networks from being employed in real-world noisy scenarios, in particular in safety critical applications such as aircraft, autonomous cars, and medical devices.

State-of-the art methods for training deep networks with noisy labels are mostly heuristics and cannot provide theoretical guarantees for the robustness of the trained model in presence of noisy labels. Failure of such systems can have a drastic effect in sensitive and safety critical applications. Our research provides a principled method for training deep networks on real-world datasets with noisy labels. Our proposed method, CRUST, is based on the recent advances in theoretical understanding of neural networks, and provides theoretical guarantee for the performance of the deep networks trained with noisy labels. We expect our method to have a far-reaching impact in deployment of deep neural networks in real-world systems. We believe our research will be beneficial for deep learning in variety of domains, and do not have any societal or ethical disadvantages.

## Acknowledgments

We gratefully acknowledge the support of DARPA under Nos. FA865018C7880 (ASED), N660011924033 (MCS); ARO under Nos. W911NF-16-1-0342 (MURI), W911NF-16-1-0171 (DURIP); NSF under Nos. OAC-1835598 (CINES), OAC-1934578 (HDR), CCF-1918940 (Expeditions), IIS-2030477 (RAPID); Stanford Data Science Initiative, Wu Tsai Neurosciences Institute, Chan Zuckerberg Biohub, Amazon, Boeing, JPMorgan Chase, Docomo, Hitachi, JD.com, KDDI, NVIDIA, Dell. J. L. is a Chan Zuckerberg Biohub investigator.

## Footnotes

[1]Code available at https://github.com/snap-stanford/crust.

[2]Note that, if $\frac{\partial\mathcal{J}(\boldsymbol{W}, \boldsymbol{X})}{\partial\boldsymbol{W}}$ is continuous, the smoothness condition holds over any compact domain (albeit for a possibly large $L$).

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
