[Supplementary Material]

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

# Appendix

Our main contribution is to show that for any dataset, clean data points cluster together in the gradient space and hence medoids of the gradients (1) have clean labels, and (2) provide a low-rank approximation of the Jacobian, $\mathcal{J}$, of an arbitrary deep network. Hence, training on the medoids is robust to noisy labels. Our analysis of the residuals during gradient descent builds on the analysis of [23], but generalize it to arbitrary deep networks without the need for over-parameterization, and random initialization. Crucially, [23] relies on the following assumptions to show that gradient descent with early stopping is robust to label noise, with a high probability: (1) data $\boldsymbol{X} \subset \mathbb{R}^{n \times d}$ has $K$ clusters, (2) neural net $f$ has one hidden layer with $k$ neurons, i.e., $f = \phi(\boldsymbol{X}\boldsymbol{W}^T)\boldsymbol{\nu}$, (3) output weights $\boldsymbol{\nu}$ are fixed to half $+1/\sqrt{k}$, and half $-1/\sqrt{k}$, (4) network is over-parameterized, i.e., $k \geq K^4$, where $K = \mathcal{O}(n)$, (5) the input-to-hidden weights $\boldsymbol{W}^0$ have random Gaussian initialization. Indeed, from the clusterable data assumption it easily follows that the neural network covariance $\Sigma(\boldsymbol{X}) = \mathbb{E}_{\boldsymbol{W}}\big[\big(\phi'(\boldsymbol{X}\boldsymbol{W}^T)\phi'(\boldsymbol{W}^T\boldsymbol{X})\big) \odot \big(\boldsymbol{X}\boldsymbol{X}^T\big)\big] = \frac{1}{k}\mathbb{E}_{\boldsymbol{W}^0}\big[\mathcal{J}(\boldsymbol{W}^0)\mathcal{J}^T(\boldsymbol{W}^0)\big]$ is low-rank, and hence early stopping prevents overfitting noisy labels. In contrast, our results holds for arbitrary deep networks without relying on the above assumptions.

# A   Proofs for Theorems

The following Corollary builds upon the meta Theorem 7.2 from [23] and captures the contraction of the residuals during gradient descent updates on a dataset with corrupted labels, when the Jacobian is exactly low-rank. The original theorem in [23] is based on the assumptions that the fraction of corrupted labels in the dataset is small, and the dataset is clusterable. Hence $\mathcal{S}_+$ is dictated by the membership of data points to different clusters. In contrast, our method CRUST applies gradient descent only to the selected subsets. Note that the elements of the selected subsets are mostly clean, and hence the extracted subsets have a significantly smaller fraction of noisy labels. The elements selected earlier by CRUST are medoids of the main clusters in the gradient space. As we keep selecting new data points, we extract medoids of the smaller groups of data points within the main clusters. Therefore, in our method the support subspace $\mathcal{S}_+$ is defined by the assignment of the elements of the extracted subsets to the main clusters.

More specifically, assume that there are $K < k$ main clusters in the gradient space. We denote the set of central elements of the main clusters by $\bar{S}$. For a subset $S \subseteq V$ of size $k$ selected by CRUST and upper-bounds $d^u$ on gradient dissimilarities from Eq. (9), let $\Lambda_\ell = \{i \in [k] | \ell = \arg\min_{s \in \bar{S}} d_{is}^u\}$ be the set of elements in $S$ that are closest to an element $\ell \in \bar{S}$, i.e., they lie within the main cluster $l$ in the gradient space. Then, $\mathcal{S}_+$ is characterized by

$$\mathcal{S}_+ = \big\{\boldsymbol{v} \in \mathbb{R}^k \,\big|\, \boldsymbol{v}_{i_1} = \boldsymbol{v}_{i_2} \quad \text{for all} \quad i_1, i_2 \in \Lambda_\ell \quad \text{and for all } 1 \leq \ell \leq K\big\}. \tag{12}$$

The following corollary captures the contraction of the residuals during gradient descent on subsets found by CRUST, when the Jacobian is low-rank. In Theorem 4.1, we characterize the contraction of the residuals, when the Jacobian is approximately low-rank, i.e., over $\mathcal{S}_-$ the spectral norm is small but nonzero.

**Corollary A.1** *Consider a nonlinear least squares problem of the form $\mathcal{L}(\boldsymbol{W}, \boldsymbol{X}) = \frac{1}{2}\|f(\boldsymbol{W}, \boldsymbol{X}) - \boldsymbol{y}\|_{\ell_2}^2$. Assume that the weighted Jacobian of the subset $S$ found by CRUST is low-rank, i.e., for all $\boldsymbol{v} \in \mathcal{S}_+$ and $\boldsymbol{w} \in \mathcal{S}_-$ with unit Euclidean norm, and $\beta \geq \alpha > 0$ we have that $\alpha \leq \|\mathcal{J}_r^T(\boldsymbol{W}, X_S)\boldsymbol{v}\|_{\ell_2} \leq \beta$ and $\|\mathcal{J}_r^T(\boldsymbol{W}, X_S)\boldsymbol{w}\|_{\ell_2} = 0$. Moreover, assume that the Jacobian mapping $\mathcal{J}(\boldsymbol{W}, X_{\boldsymbol{S}})$ associated to the nonlinear mapping $f$ is $L$-smooth, i.e., for all $\boldsymbol{W}^1, \boldsymbol{W}^2 \in \mathbb{R}^m$ we have $\|\mathcal{J}(\boldsymbol{W}^2) - \mathcal{J}(\boldsymbol{W}^1)\| \leq L\|\boldsymbol{W}^2 - \boldsymbol{W}^1\|_{\ell_2}$.[3] Also let $\boldsymbol{y}_s, \widetilde{\boldsymbol{y}}_S, \boldsymbol{e} = \boldsymbol{y}_S - \widetilde{\boldsymbol{y}}_S \in \mathbb{R}^k$ denote the corrupted and uncorrupted labels associated with the selected subset, and label corruption, respectively. Furthermore, suppose the initial residual $f(\boldsymbol{W}^0, X_S) - \widetilde{\boldsymbol{y}}_S$ with respect to the uncorrupted labels obey $f(\boldsymbol{W}^0, X_S) - \widetilde{\boldsymbol{y}}_S \in \mathcal{S}_+$. Then, iterates of gradient descent updates of the from (2) with a*

*learning rate $\eta \le \frac{1}{2\beta^2} \min\left(1, \frac{\alpha\beta}{L\|\boldsymbol{r}^0\|_{\ell_2}}\right)$, obey*

$$\|\boldsymbol{W}^\tau - \boldsymbol{W}^0\|_{\ell_2} \le \frac{4\|\boldsymbol{r}^0\|_{\ell_2}}{\alpha} = \frac{4\|f(\boldsymbol{W}^0, X_S) - \boldsymbol{y}_S\|_{\ell_2}}{\alpha}.$$

*Furthermore, if $\lambda > 0$ is a precision level obeying $\lambda \ge \|\Pi_{\mathcal{S}_+}(\boldsymbol{e})\|_{\ell_\infty}$, then running gradient descent updates of the from (2) with a learning rate $\eta \le \frac{1}{2\beta^2} \min\left(1, \frac{\alpha\beta}{L\|\boldsymbol{r}^0\|_{\ell_2}}\right)$, after $\tau \ge \frac{5}{\eta\alpha^2} \log\left(\frac{\|\boldsymbol{r}^0\|_{\ell_2}}{\lambda}\right)$ iterations, $\boldsymbol{W}^\tau$ achieves the following error bound with respect to the true labels*

$$\|f(\boldsymbol{W}^\tau, X_S) - \widetilde{\boldsymbol{y}}_S\|_{\ell_\infty} \le 2\lambda.$$

*Finally, if $\boldsymbol{e}$ has at most $s$ nonzeros and $\mathcal{S}_+$ is $\gamma$-diffused i.e., for any vector $\boldsymbol{v} \in \mathcal{S}_+$ we have $\|\boldsymbol{v}\|_{\ell_\infty} \le \sqrt{\gamma/n}\|\boldsymbol{v}\|_{\ell_2}$ for some $\gamma > 0$, then using $\lambda = \|\Pi_{\mathcal{S}_+}(\boldsymbol{e})\|_{\ell_\infty}$, we get*

$$\|f(\boldsymbol{W}^\tau, X_S) - \widetilde{\boldsymbol{y}}_S\|_{\ell_\infty} \le 2\|\Pi_{\mathcal{S}_+}(\boldsymbol{e})\|_{\ell_\infty} \le \frac{\gamma\sqrt{s}}{n}\|\boldsymbol{e}\|_{\ell_2}.$$

**Lemma A.2** *Let $\{(\boldsymbol{x}_i, y_i)\}_{i\in S}$ be the subset selected by CRUST, and $\{\tilde{y}_i\}_{i\in S}$ be the corresponding noiseless labels. Note that the elements of the selected subsets are mostly clean, but may contain a smaller fraction $\rho$ of noisy labels. Let $\mathcal{J}(\boldsymbol{W}, \boldsymbol{X}_S)$ be the Jacobian matrix corresponding to the selected subset which is rank $k$, and $\mathcal{S}_+$ be its column space. Then, the difference between noiseless and noisy labels satisfy the bound*

$$\|\Pi_{\mathcal{S}_+}(\boldsymbol{y}_S - \tilde{\boldsymbol{y}}_S)\|_{\ell_\infty} \le 2\rho.$$

**Proof** The proof is similar to that of Lemma 8.10 in [23], but using $\mathcal{S}_+$ as defined in Eq. (12). ∎

### A.1 Proof of Theorem 4.1

We first consider the case where the set $S \subseteq V$ of $k$ weighted medoids found by CRUST approximates the largest singular value of the neural network Jacobian by an error of $\epsilon = 0$. Therefore, we can apply Corollary A.3 to characterize the behavior of gradient descent on the weighted subsets found by CRUST. The following Corollary summarizes the results:

**Corollary A.3** *Assume that we apply gradient descent on the least-squares loss in Eq. (2) to train a neural network on subsets found by CRUST from a dataset with class labels $\{\nu_j\}_{j=1}^C \in [-1,1]$, label margin $\delta$. Suppose that the Jacobian mapping is $L$-smooth, and let $\alpha = \min_\tau \sqrt{r_{\min}^\tau}\sigma_{\min}(\mathcal{J}(\boldsymbol{W}^\tau, \boldsymbol{X}_{S^\tau}))$, and $\beta = \max_\tau \sqrt{r_{\max}^\tau}\|\mathcal{J}(\boldsymbol{W}^\tau, \boldsymbol{X}_{S^\tau})\|$ be the minimum and maximum singular values of the Jacobian of the subsets weighted by $\boldsymbol{r}^\tau$ found by CRUST, during $\tau$ steps of gradient descent updates. If subsets contain a fraction of $\rho \le \delta/8$ noisy labels, using step size $\eta = \frac{1}{2\beta^2} \min(1, \frac{\alpha\beta}{L\sqrt{2k}})$, after $\tau = \mathcal{O}(\frac{1}{\eta\alpha^2} \log(\frac{\sqrt{k}}{\rho}))$ iterations, the neural network classifies all the selected elements correctly.*

**Proof** Fix a vector $\boldsymbol{v}$ and let $\tilde{\boldsymbol{p}} = \mathcal{J}_r(\boldsymbol{W}, \boldsymbol{X}_S)\boldsymbol{v}$ and $\boldsymbol{p} = \mathcal{J}(\boldsymbol{W}, \boldsymbol{X}_S)\boldsymbol{v}$. Entries of $\tilde{\boldsymbol{p}}$ multiply the entries of $\boldsymbol{p}$ somewhere between $r_{\min}$ and $r_{\max}$. This establishes the upper and lower bounds on the singular values of $\mathcal{J}_r(\boldsymbol{W}, \boldsymbol{X}_S)$ over $\mathcal{S}_+$ in terms of the singular values of $\mathcal{J}(\boldsymbol{W}, \boldsymbol{X}_S)$. I.e.,

$$\sqrt{r_{\min}}\sigma_{\min}(\mathcal{J}(\boldsymbol{W}, \boldsymbol{X}_{S^*}), \mathcal{S}_+) \le \sigma_{i\in[k]}(\mathcal{J}_r(\boldsymbol{W}, \boldsymbol{X}_{S^*}), \mathcal{S}_+) \le \sqrt{r_{\max}}\|\mathcal{J}(\boldsymbol{W}, \boldsymbol{X}_{S^*})\|. \quad (13)$$

Therefore, we get that $\alpha = \min_\tau \sqrt{r_{\min}^\tau}\sigma_{\min}(\mathcal{J}(\boldsymbol{W}^\tau, \boldsymbol{X}_S^\tau)), \beta = \max_\tau \sqrt{r_{\max}^\tau}\|\mathcal{J}(\boldsymbol{W}^\tau, \boldsymbol{X}_S^\tau)\|$.

Moreover, we have $f : \mathbb{R}^d \to [-1, 1]$, and class labels $\{\nu_j\}_{j=1}^C \in [-1, 1]$. Hence, for every element $i \in S$, we have $|f(\boldsymbol{W}, \boldsymbol{x}_i) - y_i| \le 2$, and the upper-bound on the initial misfit is $\|\boldsymbol{r}^0\|_{l_2} \le \sqrt{2k}$.

Now, using Lemma A.2, we know that

$$\|\Pi_{\mathcal{S}_+}(\boldsymbol{y} - \tilde{\boldsymbol{y}})\|_{\ell_\infty} \le 2\rho.$$

Substituting the values corresponding to $\alpha, \beta$ in Theorem 4.1, we get that after

$$\frac{5}{\eta\alpha^2} \log(\frac{\|\boldsymbol{r}^0\|_{\ell_2}}{2\rho}) \le \frac{5}{\eta\alpha^2} \log(\frac{\sqrt{2k}}{2\rho}) \le \tau \quad (14)$$

gradient descent iterations, the error bound with respect to the true labels is $\|f(\boldsymbol{W}_\tau, X_S) - \widetilde{\boldsymbol{y}}_S\|_{\ell_\infty} \le 2\rho$. If gradient clusters are roughly balanced, i.e., there are $\mathcal{O}(k/K')$ data points in each cluster, we get that for all gradient iterations with

$$\frac{5}{\eta\alpha^2}\log\left(\frac{\|\boldsymbol{r}^0\|_{\ell_2}}{2\rho}\right) \le \frac{5}{\eta\alpha^2}\log\left(\frac{\sqrt{2k}}{2\rho}\right) = \mathcal{O}\left(\frac{K}{\eta k\sigma_{\min}^2 \mathcal{J}(\boldsymbol{W}, \boldsymbol{X}_S)}\log\left(\frac{\sqrt{k}}{\rho}\right)\right) \le \tau, \qquad (15)$$

the infinity norm of the residual obeys (using $\lambda = \|\Pi_{\mathcal{S}_+}(\boldsymbol{e})\|_{\ell_\infty} \le 2\rho$)

$$\|f(\boldsymbol{W}, X_S) - \tilde{\boldsymbol{y}}\|_{\ell_\infty} \le 4\rho.$$

This implies that if $\rho \le \delta/8$, the labels predicted by the network are away from the correct labels by less than $\delta/2$, hence the elements of the subsets (including noisy ones) will be classified correctly. ∎

## A.2   Completing the Proof of Theorem 4.1

Next, we consider the case where weighted subsets found by CRUST approximate the prominent singular value of the Jacobian matrix by an error of at most $\epsilon$. Here, we characterize the behavior of gradient descent by comparing the iterations with and without error.

In particular, starting from $\boldsymbol{W}^0 = \bar{\boldsymbol{W}}^0$ consider the gradient descent iterations on the weighted subsets $\bar{S}$ that estimating the largest singular value of the neural network Jacobian without an error,

$$\bar{\boldsymbol{W}}^{\tau+1} = \bar{\boldsymbol{W}}^\tau - \eta \mathcal{J}_r^T(\bar{\boldsymbol{W}}^\tau, \boldsymbol{X}_{\bar{S}^\tau})(f(\bar{\boldsymbol{W}}^\tau, \boldsymbol{X}) - \boldsymbol{y}_{\bar{S}^\tau}), \qquad (16)$$

and gradient descent iterations on the weighted subsets $S$ with an error of at most $\epsilon$ in estimating the largest singular value of the neural network Jacobian,

$$\boldsymbol{W}^{\tau+1} = \boldsymbol{W}^\tau - \eta \mathcal{J}_r^T(\boldsymbol{W}^\tau, \boldsymbol{X}_{S^\tau})(f(\boldsymbol{W}^\tau, \boldsymbol{X}_{S^\tau}) - \boldsymbol{y}_{S^\tau}). \qquad (17)$$

To proceed with the proof, we use the following short hand notations for residuals and Jacobian matrix in iteration $\tau$ of gradient descent:

$$\boldsymbol{r}^\tau = f(\boldsymbol{W}^\tau, \boldsymbol{X}_{S^\tau}) - \boldsymbol{y}_{S^\tau}, \ \bar{\boldsymbol{r}}^\tau = f(\bar{\boldsymbol{W}}^\tau, \boldsymbol{X}_{\bar{S}^\tau}) - \boldsymbol{y}_{\bar{S}^\tau} \qquad (18)$$

$$\mathcal{J}^\tau = \mathcal{J}_r(\boldsymbol{W}^\tau, \boldsymbol{X}_{S^\tau}), \ \mathcal{J}^{\tau+1,\tau} = \mathcal{J}_r(\boldsymbol{W}^{\tau+1}, \boldsymbol{W}^\tau, \boldsymbol{X}_{S^\tau}), \qquad (19)$$

$$\bar{\mathcal{J}}^\tau = \mathcal{J}_r(\bar{\boldsymbol{W}}^\tau, \boldsymbol{X}_{S^\tau}), \ \bar{\mathcal{J}}^{\tau+1,\tau} = \mathcal{J}_r(\bar{\boldsymbol{W}}^{\tau+1}, \bar{\boldsymbol{W}}^\tau, \boldsymbol{X}_{\bar{S}^\tau}) \qquad (20)$$

$$d^\tau = \|\boldsymbol{W}^\tau - \bar{\boldsymbol{W}}^\tau\|_F, \ p^\tau = \|\boldsymbol{r}^\tau - \bar{\boldsymbol{r}}^\tau\|_F, \qquad (21)$$

where $\mathcal{J}_r(\boldsymbol{W}^1, \boldsymbol{W}^2, \boldsymbol{X}_S)$ denotes the average weighted neural network Jacobian at subset $\boldsymbol{X}_S$, i.e.,

$$\mathcal{J}_r(\boldsymbol{W}^1, \boldsymbol{W}^2, \boldsymbol{X}_S) = \int_0^1 \mathcal{J}_r(\alpha\boldsymbol{W}^1 + (1-\alpha)\boldsymbol{W}^2, \boldsymbol{X}_S)d\alpha.$$

We first proof the following Lemma that bounds the normed difference between $\mathcal{J}_r(\bar{\boldsymbol{W}}^1, \bar{\boldsymbol{W}}^2, \boldsymbol{X}_{\bar{S}})$ and $\mathcal{J}_r(\boldsymbol{W}^1, \boldsymbol{W}^2, \boldsymbol{X}_S)$.

**Lemma A.4** *Let $\boldsymbol{X}_S, \boldsymbol{X}_{\bar{S}}$ be the subset of data points found by* CRUST*, that approximates the Jacobian matrix on the entire data by and error of 0 and $\epsilon$, respectively. Given parameters $\boldsymbol{W}^1, \boldsymbol{W}^2, \bar{\boldsymbol{W}}^1, \bar{\boldsymbol{W}}^2$, we have that*

$$\|\mathcal{J}_r(\boldsymbol{W}^1, \boldsymbol{W}^2, \boldsymbol{X}_S) - \mathcal{J}_r(\bar{\boldsymbol{W}}^1, \bar{\boldsymbol{W}}^2, \boldsymbol{X}_{\bar{S}})\| \le \left(\frac{\|\bar{\boldsymbol{W}}^1 - \boldsymbol{W}^1\|_F + \|\bar{\boldsymbol{W}}^2 - \boldsymbol{W}^2\|_F}{2} + \epsilon\right).$$

**Proof** Given $\boldsymbol{W}, \bar{\boldsymbol{W}}$, we can write

$$\|\mathcal{J}_r(\boldsymbol{W}, \boldsymbol{X}_S) - \mathcal{J}_r(\bar{\boldsymbol{W}}, \boldsymbol{X}_{\bar{S}})\| \le \|\mathcal{J}_r(\boldsymbol{W}, \boldsymbol{X}_S) - \mathcal{J}_r(\bar{\boldsymbol{W}}, \boldsymbol{X}_S)\| + \|\mathcal{J}_r(\bar{\boldsymbol{W}}, \boldsymbol{X}_{\bar{S}}) - \mathcal{J}_r(\bar{\boldsymbol{W}}, \boldsymbol{X}_S)\| \qquad (22)$$

$$\le L\|\boldsymbol{W} - \bar{\boldsymbol{W}}\| + \epsilon. \qquad (23)$$

To get the result on $\|\mathcal{J}(\boldsymbol{W}^1, \boldsymbol{W}^2, \boldsymbol{X}_S) - \mathcal{J}_r(\bar{\boldsymbol{W}}^1, \bar{\boldsymbol{W}}^2, \boldsymbol{X}_{\bar{S}})\|$, we integrate

$$\|\mathcal{J}(\boldsymbol{W}^1, \boldsymbol{W}^2, \boldsymbol{X}_S) - \mathcal{J}_r(\bar{\boldsymbol{W}}^1, \bar{\boldsymbol{W}}^2, \boldsymbol{X}_{\bar{S}})\| \le \int_0^1 (L\|\alpha(\bar{\boldsymbol{W}}^1 - \boldsymbol{W}^1) + (1-\alpha)(\bar{\boldsymbol{W}}^1 - \boldsymbol{W}^1)\|_F + \epsilon)d\alpha \qquad (24)$$

$$\le \frac{L(\|\bar{\boldsymbol{W}}^1 - \boldsymbol{W}^1\|_F + \|\bar{\boldsymbol{W}}^2 - \boldsymbol{W}^2\|_F)}{2} + \epsilon. \qquad (25)$$

Now, applying Lemma A.4, we have

$$\|\mathcal{J}_r(\boldsymbol{W}^\tau, \boldsymbol{X}_{S^\tau}) - \mathcal{J}_r(\bar{\boldsymbol{W}}^\tau, \boldsymbol{X}_{\bar{S}^\tau})\| \le L\|\bar{\boldsymbol{W}}^\tau - \boldsymbol{W}^\tau\|_F + \epsilon \le Ld^\tau + \epsilon \tag{26}$$

$$\|\mathcal{J}_r(\boldsymbol{W}^{\tau+1}, \boldsymbol{W}^\tau, \boldsymbol{X}_{S^\tau}) - \mathcal{J}_r(\bar{\boldsymbol{W}}^{\tau+1}, \bar{\boldsymbol{W}}^\tau, \boldsymbol{X}_{\bar{S}^\tau})\| \le L(d^\tau + d^{\tau+1})/2 + \epsilon. \tag{27}$$

Following this and since the normed noiseless residual is non-increasing and satisfies $\|\bar{\boldsymbol{r}}_\tau\|_{\ell_2} \le \|\bar{\boldsymbol{r}}^0\|_{\ell_2}$, we can write

$$\boldsymbol{W}^{\tau+1} = \boldsymbol{W}^\tau - \eta\mathcal{J}^\tau\boldsymbol{r}^\tau \quad, \quad \bar{\boldsymbol{W}}^{\tau+1} = \bar{\boldsymbol{W}}^\tau - \eta(\bar{\mathcal{J}}^\tau)^T\bar{\boldsymbol{r}}^\tau \tag{28}$$

$$\|\boldsymbol{W}^{\tau+1} - \bar{\boldsymbol{W}}^{\tau+1}\|_F \le \|\boldsymbol{W}^\tau - \bar{\boldsymbol{W}}^\tau\|_F + \eta\|\mathcal{J}^\tau - \bar{\mathcal{J}}^\tau\|\|\bar{\boldsymbol{r}}^\tau\|_{\ell_2} + \eta\|\mathcal{J}^\tau\|\|\boldsymbol{r}^\tau - \bar{\boldsymbol{r}}^\tau\|_{\ell_2}, \tag{29}$$

$$d^{\tau+1} \le d^\tau + \eta\big((Ld^\tau + \epsilon)\|\bar{\boldsymbol{r}}^0\|_{\ell_2} + \beta p^\tau\big). \tag{30}$$

For the residual we have

$$\boldsymbol{r}^{\tau+1} = \boldsymbol{r}^\tau - f(\boldsymbol{W}^\tau, \boldsymbol{X}_S) + f(\boldsymbol{W}^{\tau+1}, \boldsymbol{X}_S) \tag{31}$$

$$= \boldsymbol{r}^\tau + \mathcal{J}^{\tau+1,\tau}(\boldsymbol{W}^{\tau+1} - \boldsymbol{W}^\tau) \tag{32}$$

$$= \boldsymbol{r}^\tau - \eta\mathcal{J}^{\tau+1,\tau}(\mathcal{J}^\tau)^T\boldsymbol{r}^\tau, \tag{33}$$

where in Eq. (33) we used $\boldsymbol{W}^{\tau+1} - \boldsymbol{W}^\tau = \eta\nabla\mathcal{L}(\boldsymbol{W}^\tau) = \eta(\mathcal{J}^\tau)^T\boldsymbol{r}^\tau$. Furthermore, we can write

$$\boldsymbol{r}^{\tau+1} - \bar{\boldsymbol{r}}^{\tau+1} = (\boldsymbol{r}^\tau - \bar{\boldsymbol{r}}^\tau) - \eta(\mathcal{J}^{\tau+1,\tau} - \bar{\mathcal{J}}^{\tau+1,\tau})(\mathcal{J}^\tau)^T\boldsymbol{r}^\tau \tag{34}$$

$$- \eta\bar{\mathcal{J}}_{\tau+1,\tau}\big((\mathcal{J}^\tau)^T - (\bar{\mathcal{J}}^\tau)^T\big)\boldsymbol{r}^\tau - \eta\bar{\mathcal{J}}^{\tau+1,\tau}(\bar{\mathcal{J}}^\tau)^T(\boldsymbol{r}^\tau - \bar{\boldsymbol{r}}^\tau) \tag{35}$$

$$= \big(\boldsymbol{I} - \eta\bar{\mathcal{J}}^{\tau+1,\tau}(\bar{\mathcal{J}}^\tau)^T\big)(\boldsymbol{r}^\tau - \bar{\boldsymbol{r}}^\tau) - \eta(\mathcal{J}^{\tau+1,\tau} - \bar{\mathcal{J}}^{\tau+1,\tau})(\mathcal{J}^\tau)^T\boldsymbol{r}^\tau \tag{36}$$

$$- \eta\bar{\mathcal{J}}^{\tau+1,\tau}\big((\mathcal{J}^\tau)^T - (\bar{\mathcal{J}}^\tau)^T\big)\boldsymbol{r}^\tau. \tag{37}$$

Using $\boldsymbol{I} \succeq \bar{\mathcal{J}}^{\tau+1,\tau}(\bar{\mathcal{J}}^\tau)^T/\beta^2 \succeq 0$, we have

$$\|\boldsymbol{r}^{\tau+1} - \bar{\boldsymbol{r}}^{\tau+1}\|_{\ell_2} \le \|\boldsymbol{r}^\tau - \bar{\boldsymbol{r}}^\tau\|_{\ell_2} + \eta\beta\|\boldsymbol{r}^\tau\|_{\ell_2}(L(3d^\tau + d^{\tau+1})/2 + 2\epsilon) \tag{38}$$

$$\le \|\boldsymbol{r}^\tau - \bar{\boldsymbol{r}}^\tau\|_{\ell_2} + \eta\beta(\|\bar{\boldsymbol{r}}^0\|_{\ell_2} + p^\tau)(L(3d^\tau + d^{\tau+1})/2 + 2\epsilon), \tag{39}$$

where we used $\|\boldsymbol{r}^\tau\|_{\ell_2} \le p^\tau + \|\bar{\boldsymbol{r}}^0\|_{\ell_2}$ and $\|(\boldsymbol{I} - \eta\bar{\mathcal{J}}^{\tau+1,\tau}(\bar{\mathcal{J}}^\tau)^T)\boldsymbol{v}\|_{\ell_2} \le \|\boldsymbol{v}\|_{\ell_2}$ which follows from the contraction of the residual. This implies that

$$p^{\tau+1} \le p^\tau + \eta\beta(\|\bar{\boldsymbol{r}}^0\|_{\ell_2} + p^\tau)(L(3d^\tau + d^{\tau+1})/2 + 2\epsilon). \tag{40}$$

We use a similar inductive argument as that of Theorem 8.10 in [23]. The claim is that if for all $t \le \tau_0$, we have (using $\|\bar{\boldsymbol{r}}^0\|_{\ell_2} \le \Theta$)

$$\epsilon \le \mathcal{O}\big(\frac{\alpha^2}{\beta\log(\sqrt{k}/\rho)}\big) \le \mathcal{O}\big(\frac{k\sigma_{\min}^2(\mathcal{J}(\boldsymbol{W}, \boldsymbol{X}_S))}{K\beta\log(\sqrt{k}/\rho)}\big), \quad \text{and} \quad L \le \frac{2}{5\tau_0\eta\Theta(1 + 8\eta\tau_0\beta^2))}, \tag{41}$$

then it follows that

$$p^t \le 8t\eta\epsilon\Theta\beta, \qquad d^t \le 2t\eta\epsilon\Theta(1 + 8\eta\tau_0\beta^2) \le 20t\eta^2\tau_0\epsilon\Theta\beta^2 \le \mathcal{O}(t\eta^2\tau_0 k^{3/2}\epsilon). \tag{42}$$

The proof is by induction. Suppose for $t \le \tau_0 - 1$, we have that

$$p^t \le 8t\eta\epsilon\Theta\beta \le \Theta, \qquad d^t \le 2t\eta\epsilon\Theta(1 + 8\eta\tau_0\beta^2). \tag{43}$$

At $t + 1$, from (30) we know that

$$\frac{d^{t+1} - d^t}{\eta} \le Ld^t\Theta + \epsilon\Theta + 8\tau_0\eta\beta^2\epsilon\Theta \tag{44}$$

Now, using $L \le \frac{2}{5\tau_0\eta\Theta(1+8\eta\tau_0\beta^2))} \le \frac{1}{2\eta\tau_0\Theta}$ from (41), and replacing $d^t$ from (43) into (44) we get

$$\frac{d^{t+1} - d^t}{\eta} \le Ld^t\Theta + \epsilon\Theta + 8\tau_0\eta\beta^2\epsilon\Theta \overset{?}{\le} 2\epsilon\Theta(1 + 8\eta\tau_0\beta^2). \tag{45}$$

This establishes the induction for $d^{t+1}$.

To show the induction on $p^t$, following (8.64) and using $p^t \leq \Theta$, we need

$$\frac{p^{t+1} - p^t}{\eta} \leq \beta\Theta(L(3d^\tau + d^{\tau+1}) + 4\epsilon) \overset{?}{\leq} 8\epsilon\Theta\beta \tag{46}$$

$$L(3d^\tau + d^{\tau+1}) + 4\epsilon \overset{?}{\leq} 8\epsilon \tag{47}$$

$$L(3d^\tau + d^{\tau+1}) \overset{?}{\leq} 4\epsilon \tag{48}$$

$$10L\tau_0\eta(1 + 8\eta\tau_0\beta^2)\Theta \overset{?}{\leq} 4, \tag{49}$$

where in the last inequality we used $3d^t + d^{t+1} \leq 10\tau_0\eta\epsilon\Theta(1 + 8\eta\tau_0\beta^2)$. Note that $\eta = \frac{1}{2\beta^2}\min(1, \frac{\alpha\beta}{L\sqrt{2k}})$. Hence, if $\frac{\alpha\beta}{L\sqrt{2k}} \geq 1$, we get that $\eta = \frac{1}{2\beta^2} \geq \frac{1}{\tau_0\beta^2}$, and thus $\eta\tau_0\beta^2 \geq 1$. This allows upper-bounding $3d^t + d^{t+1}$. Now, for $L$ we have

$$L \leq \frac{2}{5\tau_0\eta\Theta(1 + 8\eta\tau_0\beta^2))}. \tag{50}$$

This concludes the induction since the condition on $L$ is satisfied.

Now, from (44) and using $\eta\tau_0\beta^2 \geq 1$ we get

$$d^t \leq 2t\eta\epsilon\Theta(1 + 8\eta\tau_0\beta^2) \leq \mathcal{O}(t\eta^2\tau_0\sqrt{k}\epsilon). \tag{51}$$

Finally, from (43) we have $p^t \leq 8t\eta\epsilon\Theta\beta \leq \Theta$. Using $\eta\tau_0 = \mathcal{O}(\frac{K}{k\sigma_{\min}^2 \mathcal{J}(\boldsymbol{W}, \boldsymbol{X}_S)} \log(\frac{\sqrt{k}}{\rho}))$ and noting that $\alpha \leq \beta$ we have that for $\tau \geq t$

$$\epsilon \leq \frac{1}{8\tau_0\eta\beta} \leq \mathcal{O}(\frac{\alpha^2}{\beta\log(\sqrt{k}/\rho)}) \leq \mathcal{O}(\frac{k\sigma_{\min}^2(\mathcal{J}(\boldsymbol{W}, \boldsymbol{X}_S))}{K\beta\log(\sqrt{k}/\rho)}).$$

Now we calculate the misclassification error. From Corollary A.3 and Eq. (43) we have that for $\eta = \frac{1}{2\beta^2}\min(1, \frac{\alpha\beta}{L\sqrt{2k}})$ and $\Theta = \sqrt{2k}$, after $\tau = \mathcal{O}(\frac{K}{\eta k\sigma_{\min}^2 \mathcal{J}(\boldsymbol{W}, \boldsymbol{X}_S)} \log(\frac{\sqrt{k}}{\rho}))$ iterations, we get

$$\|\bar{\boldsymbol{r}}^\tau\|_{\ell_\infty} \leq 4\rho \quad \text{and} \tag{52}$$

$$\|p^t\|_{\ell_2} = \|\boldsymbol{r}^\tau - \bar{\boldsymbol{r}}^\tau\|_{\ell_2} \leq c\epsilon \frac{K\beta}{\sqrt{k}\sigma_{\min}^2 \mathcal{J}(\boldsymbol{W}, \boldsymbol{X}_S)} \log(\frac{\sqrt{k}}{\rho}). \tag{53}$$

To calculate the classification rate, we denote the residual vectors $\bar{\boldsymbol{r}}^\tau = f(\bar{\boldsymbol{W}}^\tau, \boldsymbol{X}_{\bar{S}^\tau}) - \tilde{\boldsymbol{y}}_{S^\tau}$ and $\boldsymbol{r}^\tau = f(\boldsymbol{W}^\tau, \boldsymbol{X}_{S^\tau}) - \tilde{\boldsymbol{y}}_{S^\tau}$. Now, we count the number of entries of $\boldsymbol{r}^\tau$ that is larger than the label margin $\delta/2$ in absolute value. Let $\mathcal{I}$ be the set of entries satisfying this condition. For $i \in \mathcal{I}$ we have $|r_i^\tau| \geq \delta/2$. Therefore,

$$|\bar{r}_i^\tau| + |r_i^\tau - \bar{r}_i^\tau| \geq |r_i^\tau + \bar{r}_i^\tau - \bar{r}_i^\tau| \geq \delta/2, \tag{54}$$

Since $\rho = (1 - \gamma)\delta/8 < \delta/8$ for $0 < \gamma \ll 1$, we get $\|\bar{\boldsymbol{r}}^\tau\|_{\ell_\infty} \leq 4\rho \leq (1 - \gamma)\delta/2$ and

$$|r_i^\tau - \bar{r}_i^\tau| \geq \gamma\delta/2. \tag{55}$$

Thus, the sum of the entries with a larger error than the label margin is

$$\|\boldsymbol{r}^\tau - \bar{\boldsymbol{r}}^\tau\|_{\ell_1} \geq |\mathcal{I}|\gamma\delta/2. \tag{56}$$

Consequently, we have

$$|\mathcal{I}|\gamma\delta/2 \leq \|\boldsymbol{r}^\tau - \bar{\boldsymbol{r}}^\tau\|_{\ell_1} \leq \sqrt{k}\|\boldsymbol{r}^\tau - \bar{\boldsymbol{r}}^\tau\|_{\ell_2} \leq c\epsilon \frac{K\beta}{\sigma_{\min}^2 \mathcal{J}(\boldsymbol{W}, \boldsymbol{X}_S)} \log(\frac{\sqrt{k}}{\rho}). \tag{57}$$

Hence, the total number of errors is at most

$$|\mathcal{I}| \leq c'\epsilon \frac{K\beta}{\gamma\delta\sigma_{\min}^2 \mathcal{J}(\boldsymbol{W}, \boldsymbol{X}_S)} \log(\frac{\sqrt{k}}{\rho}) = \mathcal{O}(\frac{\epsilon k\beta}{\gamma\delta\alpha^2} \log(\frac{\sqrt{k}}{\rho})). \tag{58}$$

For the network to classify all the data points correctly, we need $|\mathcal{I}| \leq c'\epsilon \frac{K\beta}{\gamma\delta\sigma_{\min}^2 \mathcal{J}(\boldsymbol{W}, \boldsymbol{X}_S)} \log(\frac{\sqrt{k}}{\rho}) < 1$. Hence, we get that

$$\epsilon < \frac{c_0\gamma\delta\sigma_{\min}^2(\mathcal{J}(\boldsymbol{W}, \boldsymbol{X}_S))}{K\beta\log(\frac{\sqrt{k}}{\rho})} < \frac{c_1\delta\sigma_{\min}^2(\mathcal{J}(\boldsymbol{W}, \boldsymbol{X}_S))}{K\beta\log(\frac{\sqrt{k}}{\rho})}. \tag{59}$$