[Reviews · NeurIPS 2020]

Review 1

Summary and Contributions: The paper studies the robust training of neural networks in the presence of label noise. Since overparameterized neural networks can fit the noisy labels, naively training on corrupted data can lead to poor generalization. This paper exploits the bimodal nature of the Jacobian matrix of neural networks where the Jacobian matrix is approximately low rank with a few large singular values. Furthermore, the residual associated with the clean labels mostly lies in the subspace spanned by the singular vectors associated with the large singular values of the Jacobian matrix; and the residual corresponding to noisy labels mostly resides in the complement subspace. The paper proposes to sample a subset of training points in every iteration such that the Jacobian associated with the sampled points forms a low-rank approximation of the original Jacobian matrix. To mitigate the effect of the points with noisy labels selected in the subset, the paper proposes to utilize the mixup technique. The proposed method is validated on CIFAR-10 and CIFAR-100 with synthetically generated noisy labels and Webvision dataset with naturally occurring noisy labels. ############### Post author response phase ############### Thank you for addressing my comments, especially regarding the comparison with the prior work [19]. I have updated the score accordingly.

Strengths: There is significant scope for improvement in the presentation of the paper. There are numerous typos and many of the notations in the paper are not well defined. Please see the feedback section below. Despite the claim in Section 2 that the techniques in prior work are limited, the proofs in the paper heavily rely on the techniques developed in [19]. A more thorough discussion on the similarity and difference of the technical results and tools of [19] and this paper in the main text would help put the technical contribution of this paper in the right context. How would the proposed method behave in the settings with a very large number of labels (such as extreme classification) where class margin might not hold. Would utilizing square loss give good performance in such settings?

Weaknesses: There is significant scope for improvement in the presentation of the paper. There are are numerous typos and many of the notations in the paper are not well defined. Please see the feedback section below. Despite the claim in Section 2 that the techniques in prior work are limited, the proofs in the paper heavily rely on the techniques developed in [19]. A more thorough discussion on the similarity and difference of the technical results and tools of [19] and this paper in main text would help put the technical contribution of this paper in the right context. How would the propose method behave in the settings with very large number of labels (such as extreme classification) where class margin might not hold. Would utilizing square loss give good performance in such settings?

Correctness: As far as the reviewer can tell, the claims and methods are correct.

Clarity: The paper does a reasonable job of conveying the main ideas of the proposed methods. However, there is quite a bit of room for improvement in the presentation of the paper.

Relation to Prior Work: The paper presents a detailed account of the prior work on training in the presence of label noise. As mentioned in the weaknesses section above, there is some need for elaboration on the comparison with [19].

Reproducibility: Yes

Additional Feedback: Notation related issues and typos (not comprehensive). 1. It's not clear what the dimension of J(W, X_S) is. Eq. (5) gives the impression that it should be n x m, because of the term || J(W, X) - J(W, X_S)||. However, line 160 and Proof of Corollary B.3 make sense when it is k x m. 2. \mathcal{S}_+ and \mathcal{S}_{-} are not defined. Are these subspaces? 3. Notation \sigma_i(J(..), \mathcal{S}_{+}) is not defined. 4. Both W^{\tau} (superscript) and W_{\tau} (subscript) are used as iterates. Please consistently use the same notation. 5. In Algorithm 1 (step 6), $k/n_c$-medoids --> $kn_c$ medoids? Accordingly, fix Section 4.3. 6. In Broader Impact sections, please fix lines 325--330. 7. In line 479--480, should $f$ be replaced with $\mathcal{L}$? Other minor comments/questions: 1. In Table 2, what is the difference between 'coreset w/ label' and 'coreset w/ pred.'? 2. What is the utility of line 490 - 491 and Eq. (16). Similar to this many sections of the appendix lack coherent structure. 3. In practice, one typically employs SGD with much smaller mini-batches compared to the full dataset. Could the authors comment on the relevance of their method in such settings?


Review 2

Summary and Contributions: This paper proposes CRUST, a method to train neural networks with noisy labels. This improves on previous methods by being computationally simple and having the added benefit of theoretical performance guarantees. The main idea is to approximate the Jacobian of the paramaters wrt data with a low rank approximation by solving a k-medoids problem. This leads to several improvements over state of the art on symmetric/asymmetric synthetic noise, as well as a real-world noisy dataset (WebVision)

Strengths: Relevance/Significance: This paper provides theoretical guarantees in an area where recent progress has been largely empirical. Additionally, the method is simple and quite efficient, for example not requiring an auxiliary model Novelty: builds on previous work [26] in a novel way for a distinct application area Experiments: Ablation on most components of the algorithm

Weaknesses: Experiments: In addition to ablation on mixup and coresets with label vs prediction, it would be good to compare the method with a standard low rank approximation in place of the coverage objective (e.g. QR decomposition or the references below)

Correctness: Proofs appear to be correct

Clarity: The paper flows well and is generally a pleasure to read. However, the authors should clarify they propose a theoretical argument (not a rigorous proof such as Theorem 4.1) that mixup improves coreset quality and therefore leads to better generalization Typo: "point to its closes medoid" should be "closest medoid"

Relation to Prior Work: This paper has extensive discussion of training neural networks subject to label noise, but there is less mention of work on low rank matrix approximation/column subset selection e.g. [1-3]. Additionally, the authors should discuss similarity to [4] which in some settings (CIFAR-10, WebVisioin) achieves similar error reductions without mixup while meeting several of the desiderata proposed here. [1] Cohen et al. Input Sparsity Time Low-Rank Approximation via Ridge Leverage Score Sampling. SODA 2017 [2] Altschuler et al. Greedy Column Subset Selection: New Bounds and Distributed Algorithms. ICML 2016 [3] Khanna et al. On Approximation Guarantees for Greedy Low Rank Optimization. ICML 2017 [4] Pleiss et al. Identifying Mislabeled Data using the Area Under the Margin Ranking. https://arxiv.org/abs/2001.10528

Reproducibility: Yes

Additional Feedback: I think this is paper presents a good, novel idea in a relevant area. However, there are several questions wrt experimental setup and related work which should be resolved Questions: - Why not minimize the rank directly, or compare to other approximate low rank objectives mentioned above? - Does the proposed method scale to TinyImageNet or the full WebvVision/ImageNet datasets? - How does solving (5) compare to using the best low-rank subspace or best subset of columns? - If k is too large, will CRUST overfit to noisy labels? - How do the running times of proposed algorithms compare, and is there significant speedup from training on a subset of the data? - Do the 30-70% of data removed during selection correspond to data corrupted by noisy labels? It would be good to analyze how clean the coreset is EDIT: After reading the author feedback I am keeping my review the same


Review 3

Summary and Contributions: This paper addresses the problem of learning with label noise. The authors propose to iteratively select clean data points that provide an approximately low-rank Jacobian matrix, which avoids overfitting caused by label noise. The proposed method is strongly theoretically grounded and makes a significant contribution to label noise learning community. Experimental results on multiple benchmark datasets show that CRUST outperforms the state-of-the-art methods. **After reading author response** I have read the rebuttal carefully, and I will not change my score.

Strengths: 1. The idea is well motivated and clearly presented. 2. The approach has detailed and strong theoretical guarantees for coping with label noise. 3. The results on synthetic and real-world datasets demonstrate a clear improvement on the SoA. The implementation details of the proposed method are also provided, which will be very helpful in reproducing the reported results.

Weaknesses: 1. Compared with strong theoretical analysis, the experimental analysis is lacking slightly. I hope that the authors can add some baselines ([1], [2], [3]) to further verify the effectiveness of the method experimentally. 2. Noise level is low. I wonder what the performance would be in extreme cases with even more label noise. [1] Yilun Xu et al. L_DMI: A Novel Information-theoretic Loss Function for Training Deep Nets Robust to Label Noise. NeurIPS 2019. [2] Xiaobo Xia et al. Are Anchor Points Really Indispensable in Label-Noise Learning. NeurIPS 2019. [3] Daiki Tanaka et al. Joint optimization framework for learning with noisy labels. CVPR 2018.

Correctness: Yes, the claims, method and empirical methodology are correct.

Clarity: Yes, the paper is well written.

Relation to Prior Work: Yes, the paper is clearly discussed how this work differs from previous contributions.

Reproducibility: Yes

Additional Feedback:


Review 4

Summary and Contributions: The authors proposed a novel approach with strong theoretical guarantees for robust training of neural networks trained with noisy labels. The authors focused on selecting subsets of clean data that provide an approximately low-rank Jacobian matrix. The authors proved that gradient descent applied to the subsets cannot overfit the noisy labels, without regularization or early stopping.

Strengths: - The idea to leverage the low-rank Jacobian matrix to alleviate label noise seems novel. - The authors provided theoretical proof of the advantages of the proposed method. - The authors provided an ablation study to demonstrate the effectiveness of each component and achieved state-of-the-art performance with the final method. - The proposed method, CRUST, improves the image classification performance noticeably.

Weaknesses: - It would be better to evaluate the proposed model on other real-world datasets with noisy labels such as the Clothing1M dataset [1]. [1] Tong Xiao et al., Learning from massive noisy labeled data for image classification. CVPR 2015

Correctness: It seems the claims, method, and empirical methodology are correct.

Clarity: The paper is well written and the motivation for the proposed framework is clear to understand.

Relation to Prior Work: It seems the paper clearly referred and discussed the difference from previous methods.

Reproducibility: Yes

Additional Feedback: I have no other comments to say. ######################### Comments after rebuttal After reading the author's responses and the other reviewers' comments, I decided to keep my decision for accept.

[Author Response · NeurIPS 2020]

We thank all the reviewers for carefully checking the paper and acknowledging the "efficiency and practicality" of our work, and that it "provides theoretical guarantees in an area where recent progress has been largely empirical". Below, we provide clarifications on technical contributions (R1), baseline comparisons under extreme noise (R2, R3), comparison to low-rank approximation (R2), and properties of coresets (R2). We will also clarify in the revised version.

**Technical contribution.** R1 asks for discussion of similarity and difference of technical results to [19]. Our main contribution is to show that for any dataset, clean data points cluster together in the gradient space and hence medoids of the gradients (1) have clean labels, and (2) provide a low-rank approximation of the Jacobian, $\mathcal{J}$, of an arbitrary deep net. Hence, training on the medoids is robust to noisy labels. In contrast, [19] relies on the following assumptions to show that gradient descent *with early stopping* is robust to label noise, with a high probability: (1) data $\boldsymbol{X} \subset \mathbb{R}^{n \times d}$ has $K$ clusters, (2) neural net $f$ has one hidden layer with $k$ neurons, i.e., $f = \phi(\boldsymbol{X}\boldsymbol{W}^T)\boldsymbol{\nu}$, (3) output weights $\boldsymbol{\nu}$ are fixed to half $+1/\sqrt{k}$, and half $-1/\sqrt{k}$, (4) network is over-parameterized, i.e., $k \geq K^4$, where $K = \mathcal{O}(n)$, (5) the input-to-hidden weights $\boldsymbol{W}_0$ have random Gaussian initialization. Crucially, from the clusterable data assumption it easily follows that the neural network covariance $\Sigma(\boldsymbol{X}) = \mathbb{E}\big[\big(\phi'(\boldsymbol{X}\boldsymbol{W}^T)\phi'(\boldsymbol{W}^T\boldsymbol{X})\big) \odot \big(\boldsymbol{X}\boldsymbol{X}^T\big)\big] = \frac{1}{k}\mathbb{E}_{\boldsymbol{W}_0}\big[\mathcal{J}(\boldsymbol{W}_0)\mathcal{J}^T(\boldsymbol{W}_0)\big]$ is low-rank, and hence early stopping prevents overfitting noisy labels. While our analysis of the residuals during gradient descent is similar to [19], our results holds for arbitrary deep nets *without early stopping or the above assumptions*.

**More baselines and higher label noise.** R3 asked for comparison to other baselines ([1, 2, 3]), and in extreme cases with even more label noise. The following table compares the average validation accuracy (5 runs) of CRUST compared to L_DMI, T-Revision ([1, 2]) on CIFAR-10 with 20%, 50%, 80% symmetric and 40% asymmetric noisy labels. We see that CRUST outperforms other baselines and shows a significant improvement under severe 80% noise.

| Noise Type | Sym | | | Asym |
|---|---|---|---|---|
| Noise Ratio | 20 | 50 | 80 | 40 |
| L_DMI | $84.3 \pm 0.4$ | $78.8 \pm 0.5$ | $36.2 \pm 1.6$ | $84.8 \pm 0.7$ |
| T-Revision | $79.3 \pm 0.5$ | $78.5 \pm 0.6$ | $20.9 \pm 2.2$ | $76.3 \pm 0.8$ |
| CRUST | $\mathbf{91.1 \pm 0.2}$ | $\mathbf{86.3 \pm 0.3}$ | $\mathbf{58.3 \pm 1.8}$ | $\mathbf{88.8 \pm 0.4}$ |

**Low-rank approximation vs. CRUST.** Minimizing the rank directly as mentioned by R2 ([1, 2, 3]) is impractical for deep nets. Indeed, Eq. 5 finds the best subset of $k$ columns from the Jacobian matrix. However, as discussed in lines 138-145, calculating the Jacobian matrix requires backpropagation on the entire dataset which is very expensive for deep nets. Moreover, finding the best subset of $k$ columns from $\mathcal{J} \in \mathbb{R}^{n \times m}$ has a complexity of $\mathrm{poly}(m,n,k)$, where $n, m$ are the number of data points and parameters in the net. Since the subset should be updated at every iteration, the complexity of the above methods becomes prohibitive for deep nets trained on large datasets. Most importantly, while this approach prohibits overfitting noisy labels, it does not help identifying the clean data points. We implemented the greedy column selection of [2] with sketching ($\delta = 0.3, \epsilon = 0.5$) and lazy evaluation ($\delta = 0.5$) on the partial derivatives w.r.t. the last layer (as upper-bounds), but could not finish training CIFAR10 due to the prohibitive running time. We thank R2 for pointing out the related work on low-rank approximation, and will add the discussion to revised version.

**Scalability of CRUST.** R2 asks if CRUSTcan scale to TinyImageNet/full WebvVision/ImageNet. CRUST uses a greedy algorithm to find medoids of each class in the gradient space. The complexity of the greedy algorithm is $\mathcal{O}(nk)$. However, its complexity can be reduced to $\mathcal{O}(n)$ using stochastic methods [25], and can be further improved using lazy evaluation [24] and distributed implementations [27]. Note that this complexity does not involve any backpropagation as we use the upper-bounds calculated in Eq. (9). Hence, the subsets can be found very efficiently in parallel from all the classes, and CRUST can easily scale to large datasets with tens of millions of data points. We will add experiments on larger datasets, such as full WebvVision/ImageNet/Clothing1M suggested by R2 and R4 to the final version.

**Fraction of clean data points in coreset.** R2 asks how clean the coreset is. Fig 1 (a) shows the fraction of clean data points in subsets of size 30%, 50%, and 70% selected by CRUST in presence of 50% noisy labels. It can be seen that CRUST could successfully identify almost 95% of the clean data points, and remove the data corrupted by noisy labels.

**Speedup of training on coresets.** R2 rightly points out the speedup obtained from training on subsets found by CRUST. In fact, CRUST finds the subsets very efficiently, and training on subsets is much faster than on the entire data.

**Squared loss for extreme classification.** R1 asks if squared loss performs well when we have a large number of labels. The key idea of our approach is to find representative data points with clean labels that provide an approximately low-rank Jacobian. If every class has a small number of clean data points, they cluster closely in the gradient space and CRUST will be able to extract clean representative data points. Note that learning from noisy labels would be impossible without assuming a small number of clean data points from every class. At the same time, having a very large number of labels requires selecting larger subsets that contain representative clean data points from all the classes. We note that the use of squared loss is to facilitate the theoretical analysis and our experiments are done using a cross-entropy loss.

**Size of the coreset.** R2 asks if CRUST overfit to noisy labels when $k$ is too large. Ideally, $k$ shouldn't be larger than the number of clean data points. In the limit when we select all the data points, we overfit to noisy labels.

**Notations.** R2 asks for clarification on the following notations: We fix line 160 as $\mathcal{J}_r(\boldsymbol{W}, \boldsymbol{X}_{S^*}) = \mathrm{diag}([r_1, \cdots, r_k, 0, \cdots 0])\mathcal{J}(\boldsymbol{W}, \boldsymbol{X}_{S^*}) \in \mathbb{R}^{n \times m}$. $\sigma_i(\mathcal{J}(\boldsymbol{W}, \boldsymbol{X}), \mathcal{S}_+)$ is the $i$-th singular value of the Jacobian projected over the support subspace $\mathcal{S}_+$. In Table 2, coreset w/label and w/preds. correspond to finding coresets separately from every class based on their noisy labels, or labels predicted by the model. We thank R2 and will clarify the notations.

[Meta-Review · NeurIPS 2020]

The paper attempts to improve robustness of neural network training. Specifically, authors try to mitigate poor generalization that can arise from naively training overparameterized neural networks on noisy data/labels. The authors makes interesting observation about the Jacobian matrix of neural networks: it is low rank approximately with a few large singular values. It is further claimed that error for clean labels mostly lies in the subspace corresponding to theses large singular values; and for noisy labels, it is in the complement subspace. Authors leverage this observation to propose a training method, called CRUST, to sample a subset of training points in every iteration such that the Jacobian associated with the sampled points forms a low-rank approximation of the original Jacobian matrix. To mitigate the effect of the points with noisy labels selected in the subset, the paper proposes to utilize the mixup technique. The proposed method shows strong empirical performance. Reviewers found low rank Jacobian observation to be novel. Furthermore, theoretical analysis (which significantly relaxes the assumption in [19]) are quite interesting. Overall, the reviewers reached a consensus to accept the paper and thus I am happy to recommend an acceptance to NeurIPS. However, authors should thoroughly revision of the manuscript for typographical and grammatical errors. Also add the comparisons from rebuttal to main paper.